# Tryparedoxin peroxidase-deficiency commits trypanosomes to ferroptosis-type cell death

Marta Bogacz, R Luise Krauth-Siegel*

Biochemie-Zentrum der Universität Heidelberg, Heidelberg, Germany

**Abstract** Tryparedoxin peroxidases, distant relatives of glutathione peroxidase 4 in higher eukaryotes, are responsible for the detoxification of lipid-derived hydroperoxides in African trypanosomes. The lethal phenotype of procyclic *Trypanosoma brucei* that lack the enzymes fulfils all criteria defining a form of regulated cell death termed ferroptosis. Viability of the parasites is preserved by α-tocopherol, ferrostatin-1, liproxstatin-1 and deferoxamine. Without protecting agent, the cells display, primarily mitochondrial, lipid peroxidation, loss of the mitochondrial membrane potential and ATP depletion. Sensors for mitochondrial oxidants and chelatable iron as well as overexpression of a mitochondrial iron-superoxide dismutase attenuate the cell death. Electron microscopy revealed mitochondrial matrix condensation and enlarged cristae. The peroxidase-deficient parasites are subject to lethal iron-induced lipid peroxidation that probably originates at the inner mitochondrial membrane. Taken together, ferroptosis is an ancient cell death program that can occur at individual subcellular membranes and is counterbalanced by evolutionary distant thiol peroxidases.
DOI: https://doi.org/10.7554/eLife.37503.001

## Introduction

Ferroptosis is characterized by the iron-dependent accumulation of cellular lipid hydroperoxides to lethal levels. This form of regulated cell death has been implicated in the pathology of degenerative diseases (e.g. Alzheimer's, Huntington's and Parkinson's diseases), cancer, ischemia-reperfusion injury and kidney degeneration (for recent reviews see [*Doll and Conrad, 2017*; *Stockwell et al., 2017*; *Galluzzi et al., 2018*]). The term ferroptosis was coined in 2012 to describe a form of death in cancer cells induced by the small molecule erastin (*Dixon et al., 2012*; *Dixon et al., 2014*; *Cao and Dixon, 2016*). Consequence is an inhibition of glutathione biosynthesis and thus of glutathione peroxidase 4 (GPx4) activity. Of the eight glutathione peroxidases in mammalian cells, GPx4 is the only one that is able to detoxify lipid hydroperoxides even within membranes (*Thomas et al., 1990*; *Seiler et al., 2008*; *Brigelius-Flohé and Maiorino, 2013*). Inactivation of GPx4 by chemicals like for example RSL3 induces ferroptosis (*Yang et al., 2014*); and inducible GPx4-deletion in mice revealed that ferroptosis is a pervasive and dynamic form of cell death also in non-transformed cells (*Friedmann Angeli et al., 2014*). Heat-stressed plants can also undergo a ferroptosis-like cell death (*Distéfano et al., 2017*). A hallmark of a cell death occurring exclusively by ferroptosis is its suppression by iron chelators (*Dixon et al., 2012*) or lipophilic antioxidants (*Seiler et al., 2008*; *Krainz et al., 2016*) as well as by the depletion of polyunsaturated fatty acids in membrane lipids (*Doll et al., 2017*; *Kagan et al., 2017*).

African trypanosomes (*Trypanosoma brucei* species) are the causative agents of human sleeping sickness and Nagana cattle disease. The obligate free living protozoan parasites multiply as bloodstream (BS) form in the mammalian host and as procyclic (PC) form in the tsetse fly vector. BS *T. brucei* rely exclusively on glycolysis for energy supply and have an only rudimentary mitochondrion,

*For correspondence:
luise.krauth-siegel@bzh.uni-heidelberg.de

Competing interests: The authors declare that no competing interests exist.

**eLife digest** Plants, animals and fungi all belong to a group of organisms known as eukaryotes. Their cells host a variety of compartments, with each having a specific role. For example, mitochondria are tasked with providing the energy that powers most of the processes that keep the cell alive. Membranes delimit these compartments, as well as the cells themselves.

Iron is an element needed for chemical reactions that are essential for the cell to survive. Yet, the byproducts of these reactions can damage – 'oxidize' – the lipid molecules that form the cell's membranes, including the one around mitochondria. Unless enzymes known as peroxidases come to repair the oxidized lipids, the cell dies in a process called ferroptosis. Scientists know that this death mechanism is programmed into the cells of humans and other complex eukaryotes.

However, Bogacz and Krauth-Siegel wanted to know if ferroptosis also exists in creatures that appeared early in the evolution of eukaryotes, such as the trypanosome *Trypanosoma brucei*. This single-cell parasite causes sleeping sickness in humans and a disease called nagana in horses and cattle. Before it infects a mammal, *T. brucei* goes through an 'insect stage' where it lives in the tsetse fly; there, it relies on its mitochondrion to produce energy.

Bogacz and Krauth-Siegel now show that if the parasites in the insect stage do not have a specific type of peroxidases, they die within a few hours. In particular, problems in the membranes of the mitochondrion stop the compartment from working properly. These peroxidases-free trypanosomes fare better if they are exposed to molecules that prevent iron from taking part in the reactions that can harm lipids. They also survive more if they are forced to create large amounts of an enzyme that relies on iron to protect the mitochondrion against oxidation. Finally, using drugs that prevent ferroptosis in human cells completely rescues these trypanosomes. Taken together, the results suggest that ferroptosis is an ancient cell death program which exists in *T. brucei*; and that, in the insect stage of the parasite's life cycle, this process first damages the mitochondrion.

This last finding could be particularly relevant because the role of mitochondria in ferroptosis in mammals is highly debated. Yet, most of the research is done in cells that do not rely on this cellular compartment to get their energy. During their life cycle, trypanosomes are either dependent on their mitochondria, or they can find their energy through other sources: this could make them a good organism in which to dissect the precise mechanisms of ferroptosis.

DOI: https://doi.org/10.7554/eLife.37503.002

whereas in the PC stage, the single mitochondrion is fully elaborated and the parasites gain ATP via oxidative phosphorylation. Trypanosomes have an unusual thiol redox metabolism that is based on trypanothione [$N^1$, $N^8$-bis(glutathionyl)spermidine, $T(SH)_2$)] and the flavoenzyme trypanothione reductase (TR) (**Krauth-Siegel and Leroux, 2012**; **Manta et al., 2013**; **Manta et al., 2018**). The trypanothione system delivers the reducing equivalents for a variety of crucial pathways. Most of the reactions are mediated by tryparedoxin (Tpx), an essential distant member of the thioredoxin protein family (**Comini et al., 2007**). Trypanosomes lack catalase. Hydroperoxides are detoxified by 2-Cys-peroxiredoxins (**Tetaud et al., 2001**; **Budde et al., 2003**; **Wilkinson et al., 2003**) and non-selenium glutathione peroxidase-type (Px) enzymes (**Hillebrand et al., 2003**; **Wilkinson et al., 2003**; **Schlecker et al., 2005**). Whereas the peroxiredoxins use hydrogen peroxide and peroxynitrite as main substrates (**Thomson et al., 2003**; **Trujillo et al., 2004**), the Px-type enzymes preferably detoxify lipid-derived hydroperoxides (**Diechtierow and Krauth-Siegel, 2011**). With NADPH as ultimate electron donor, the reducing equivalents flow via TR, $T(SH)_2$, and tryparedoxin onto the peroxidases which therefore have been named tryparedoxin peroxidases (**Castro and Tomás, 2008**; **Krauth-Siegel and Comini, 2008**; **Krauth-Siegel and Leroux, 2012**; **Manta et al., 2013**).

Tandemly arranged genes encode three virtually identical Px-type enzymes (Px I, II, and III) in *T. brucei* (**Hillebrand et al., 2003**). RNA-interference against the Px-type enzymes results in a severe growth defect in both BS and PC *T. brucei* (**Wilkinson et al., 2003**; **Schlecker et al., 2005**). Proliferation of the Px-depleted BS parasites can, however, be restored by supplementing the medium with the vitamin E-analog Trolox [(±)−6-hydroxy-2,5,7,8-tetramethylchromane-2-carboxylic acid] (**Diechtierow and Krauth-Siegel, 2011**). The same is true for cells lacking GPx4, the closest related enzyme in mammals (**Seiler et al., 2008**). Selective knockout (KO) of the gene encoding the

mitochondrial Px III has only a mild and transient effect on the in vitro proliferation of BS *T. brucei* and the mutant parasites are fully infectious in the mouse model. In contrast, cells that lack the cytosolic peroxidases Px I and II die after transfer into Trolox-free medium (*Diechtierow and Krauth-Siegel, 2011*; *Hiller et al., 2014*). The lethal phenotype of the BS parasites starts at the lysosome and is closely linked to the endocytosis of iron-loaded host transferrin (*Hiller et al., 2014*).

In the PC insect stage of *T. brucei,* the presence of either the cytosolic or the mitochondrial peroxidases is sufficient for viability. However, deletion of the complete peroxidase (*px I-III*) locus is lethal. The cell death can be reverted by exogenous Trolox, deferoxamine (Dfx) or MitoQ which suggested an iron-induced lipid peroxidation that may start at the mitochondrion (*Schaffroth et al., 2016*). Here we show that the death program in the peroxidase-deficient PC *T. brucei* closely resembles ferroptosis in mammalian cells. Integrity and viability of the parasites are preserved by known ferroptosis inhibitors. Strikingly, sensors for mitochondrial oxidants and chelatable iron as well as overexpression of an iron-dependent superoxide dismutase in the mitochondrion attenuate cell lysis. Transmission electron microscopy of the Px I-III knockout (KO) cells reveals mitochondrial matrix condensation and formation of enlarged cristae. Our data show that oxidants and iron in the mitochondrial matrix play a critical role in the process and that ferroptosis is an evolutionary ancient cell death program that is prevented by distant thiol peroxidases.

## Results

### Ferroptosis inhibitors protect Px I-III-deficient PC *T. brucei* from cell death

The Px-type tryparedoxin peroxidases protect African trypanosomes against fatal membrane damages (*Diechtierow and Krauth-Siegel, 2011*; *Hiller et al., 2014*; *Schaffroth et al., 2016*). In PC *T. brucei,* deletion of the *px I-III* locus is lethal but the mutant cells proliferate like wildtype parasites if the medium is supplemented with either 100 µM of Trolox or Dfx (*Schaffroth et al., 2016*). The minimum concentration to ensure full viability of the Px I-III KO cells is 50 µM Trolox and 100 µM Dfx, respectively. 10 µM Trolox and 20 µM Dfx alone are not sufficient but in combination, restore cell viability (*Figure 1—figure supplement 1*). Evidently, partial complexation of iron by Dfx lowers the concentration of Trolox that is required to protect the cells. The concerted action of Dfx and Trolox is probably related to the fact that free iron ions react with hydroperoxides generating highly deleterious alkoxy radicals. The protective effect of radical trapping agents is usually associated with their reactivity towards peroxy radicals (*Shah et al., 2018*). Yet, this reaction would regenerate hydroperoxides and thus, in peroxidase-deficient cells maintain the vicious cycle. As shown recently, Trolox is a more powerful scavenger of alkoxy radicals compared to peroxy radicals (*Alberto et al., 2013*). The reaction converts alkoxy radicals into the respective alcohols which more likely explains the protecting effect of Trolox and probably other radical trapping agents as well.

At 1 µM, α-Tocopherol (α-Toc) fully reverts the lethal phenotype of mammalian cells lacking GPx4 (*Seiler et al., 2008*). Therefore, we asked if the biologically most active form of vitamin E is also more efficient than Trolox in protecting the Px I-III KO cells. Unexpectedly, even 100 µM α-Toc was not sufficient to fully prevent cell lysis. In a second approach, the cells were pre-loaded with 10 µM α-Toc in the presence of 100 µM Trolox and then transferred into Trolox-free medium ± α-Toc. In this case, the Px I-III KO cells even slightly proliferated within the 6 hr observation time, remarkably also in the absence of further exogenous α-Toc. Apparently, uptake of α-Toc by the parasites is slow compared to Trolox, but after proper incorporation, indeed α-Toc is the superior protecting agent. If this is due to the ability of α-Toc to integrate in the phospholipid bilayer of biological membranes is not clear. As reported recently, an analog in which the isoprenoid side chain is truncated to a methyl group proved to be an even better antioxidant than α-Toc (*Zilka et al., 2017*).

The finding that lysis of the Px I-III KO *T. brucei* was prevented by either a lipophilic antioxidant or an iron chelator was reminiscent of ferroptosis (*Dixon et al., 2012*; *Friedmann Angeli et al., 2014*; *Doll and Conrad, 2017*). Most potent inhibitors of this regulated cell death described so far are ferrostatin-1 (Fer-1) and liproxstatin-1 (Lpx-1) which protect mammalian cells at nanomolar concentrations. Indeed also in trypanosomes, 100 nM Fer-1 or 200 nM Lpx-1 completely abolished the lethality of the Px I-III KO cells (*Figure 1A*). Mechanistically, the compounds act as radical-trapping

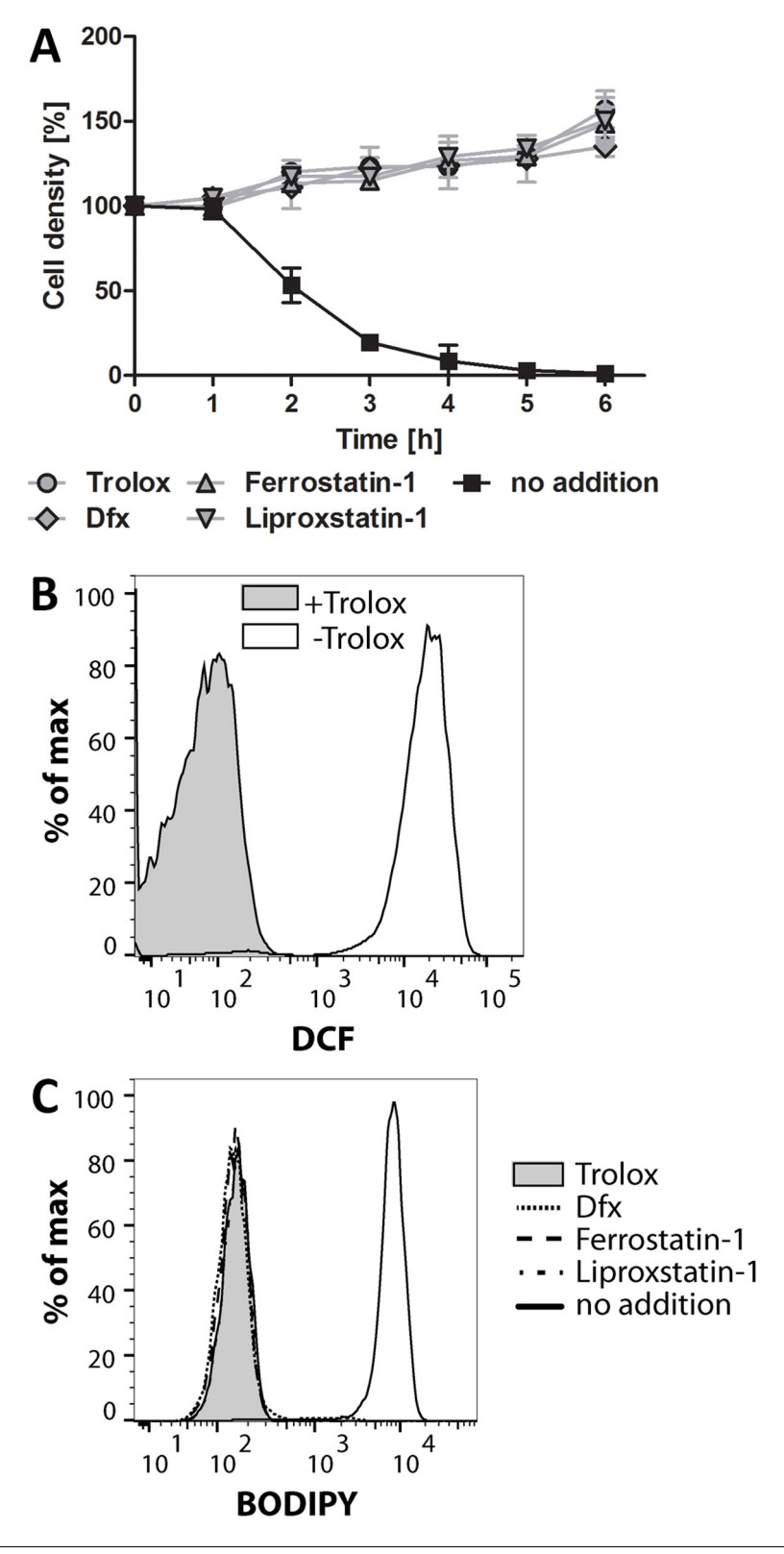

**Figure 1.** Ferroptosis inhibitors protect the Px I-III KO cells. The mutant cells were either kept in Trolox-supplemented medium or transferred into medium containing 100 µM Dfx, 100 nM ferrostatin-1 or 200 nM liproxstatin-1 or no addition. (A) Every hour viable cells were counted. The data are the mean ± SD of three independent experiments. (B) The cells were kept for 2 hr in medium ± Trolox, incubated for 30 min with 10 µM

*Figure 1 continued on next page*

*Figure 1 continued*

H$_2$DCFDA in medium + Trolox and subjected to flow cytometry measuring DCF fluorescence. (**C**) BODIPY was added and after 2 hr incubation the cells were analyzed by flow cytometry. (**B and C**) show representative histograms from two independent experiments.

DOI: https://doi.org/10.7554/eLife.37503.003

The following figure supplement is available for figure 1:

**Figure supplement 1.** Protection of Px I-III KO cells by Trolox, Dfx and α-Toc.

DOI: https://doi.org/10.7554/eLife.37503.004

agents which in phospholipid bilayers are significantly more potent antioxidants than α-Toc (*Zilka et al., 2017*).

Mammalian ferroptotic cells show a time-dependent increase of both lipid and soluble reactive oxygen species (ROS) production (*Dixon et al., 2012*; *Yang et al., 2014*). Thus, the Px I-III KO cells were incubated in medium ± Trolox and treated with H$_2$DCFDA, a broadly used dye for detection of cellular oxidants (*Kuznetsov et al., 2011*). After 2 hr in Trolox-free medium, DCF fluorescence was strongly increased indicating that the parasites are subject to general oxidative stress (*Figure 1B*). Next we measured lipid peroxidation using BODIPY 581/591 C11, a redox-sensitive dye that integrates into membranes and shifts its fluorescence from red to green upon oxidation (*Pap et al., 1999*). Px I-III KO parasites were incubated with BODIPY in medium that was supplemented with Trolox, Dfx, Fer-1 and Lpx-1, respectively, or lacked any protecting agent and analyzed by flow cytometry. Cells that were kept for 2 hr in non-supplemented medium showed strong green fluorescence whereas cells incubated in the presence of the radical scavengers or iron chelator were protected from lipid peroxidation (*Figure 1C*). Taken together, in trypanosomes, a cell death that is due to impaired removal of lipid-derived hydroperoxides can be prevented by known ferroptosis inhibitors.

## RSL3 has trypanocidal activity and inactivates Tpx

RSL3 (RAS-selective lethal) is a ferroptosis-inducing agent. The (*1S, 3R*)-RSL3 isomer kills RAS-transformed tumorigenic fibroblast cell lines with EC$_{50}$-values of 10 nM and irreversibly inactivates GPx4. The other three diastereomers display EC$_{50}$-values of 2.5 to 5 μM (*Yang et al., 2014*). BS *T. brucei* were cultured for up to 72 hr in the presence of different concentrations of (*1S, 3R*)-RSL3 or the racemic mixture and then subjected to ATPlite measurements. After 24 and 48 hr, EC$_{50}$-values of 2 and 4 μM were obtained which increased about two-fold after 72 hr probably because of long-term instability of the compounds in the medium. Thus, RSL3 was trypanocidal but selectivity for the (*1S,3R*)-isomer could not be detected (*Supplementary file 1*). A lack of stereo-specificity and EC-values in the low micromolar range were reported for non-transformed fibroblast cell lines as well (*Yang et al., 2014*). To further assess a putative interaction of RSL3 with the peroxidases, the analysis was conducted in the presence of Trolox, Lpx-1 or Fer-1. Under these conditions, viability and proliferation of *T. brucei* is independent of the Px-type enzymes (*Figure 1A*). The EC$_{50}$-values obtained in the presence or absence of the radical-trapping antioxidants were virtually the same which implies that the trypanocidal activity of RSL3 is not related to inhibition of the peroxidases. This may, at least partially, be due to the fact that the parasite peroxidases have an active site cysteine instead of the selenocysteine in GPx4. Mammalian mouse embryonic fibroblasts (MEF and PFa1 cells) in which authentic GPx4 is replaced by a Cys-mutant are much less sensitive towards (*1S,3R*)-RSL3 than the respective wildtype cells (*Ingold et al., 2018*).

RSL3 carries a chloroacetamido group. In a large scale screen against the peroxidase cascade of *T. brucei*, several compounds with this substituent proved to be trypanocidal and to inactivate Tpx (*Fueller et al., 2012*). All protein components of the parasite peroxidase system, TR, Tpx and the Px-type enzymes, are essential and possess reactive cysteine residues. Firstly, RSL3 was studied as putative covalent inhibitor of TR. 1 μM reduced TR was incubated for up to 2 hr with 100 μM RSL3 racemate as described in Materials and methods. The activity of TR remained constant ruling out any inactivation of the reductase. Next, the effect of RSL3 on the peroxidase cascade was studied. The mixture of NADPH, T(SH)$_2$, TR, Tpx and Px was treated with RSL3. After different times, H$_2$O$_2$ was added and NADPH consumption was followed. Finally, a mixture containing all components except the peroxidase was incubated with RSL3 and the reaction started by adding Px and H$_2$O$_2$. In

both approaches, RSL3 caused a time-dependent decrease of the activity but the degree of inactivation was identical irrespective of the presence or absence of the peroxidase in the pre-incubation mixture (*Supplementary file 1*). This strongly suggested that RSL3 is a time-dependent inhibitor of Tpx. The parasite-specific essential oxidoreductase is a distant relative of thioredoxins and glutaredoxins (*Comini et al., 2007*). Tpx transfers reducing equivalents from trypanothione not only to the peroxidases but also methionine-sulfoxide reductase and, probably most importantly, ribonucleotide reductase (*Dormeyer et al., 2001*). Thus, inhibition of Tpx likely affects the synthesis of DNA precursors which may be the main reason for the trypanocidal action of RSL3.

## The Px I-III-deficient cells encounter mitochondria-specific membrane damages

The PC Px I-III KO cells were incubated in medium ± Trolox, treated with the mitochondrial membrane potential-sensitive MitoTracker Red or propidium iodide (PI), as indicator of plasma membrane integrity, and subjected to flow cytometry. When kept for 4 hr in the presence of Trolox or directly after transfer into standard medium (0 hr – Trolox), the majority of cells displayed normal forward scatter (FSC) and side scatter (SSC) (*Figure 2—figure supplement 1*). After 1 or 2 hr in the absence of a protecting agent, most of the cells had reduced SSC and slightly increased FSC probably due to their reduced motility and altered morphology. From 2 hr onwards, a third population arose that comprised severely damaged or dead cells. In accordance with previous fluorescence microscopy studies (*Schaffroth et al., 2016*), already 1 hr after Trolox-withdrawal, the MitoTracker Red signal was reduced and after 2 hr had reached the minimal value. In contrast, the PI fluorescence remained at the basal level. Only when the cells were kept for ≥3 hr without the antioxidant the PI staining strongly increased. Taken together, in the Px I-III-deficient cells, loss of the mitochondrial membrane potential clearly precedes plasma membrane disintegration.

To elucidate if/how the loss of the mitochondrial membrane potential affects the morphology of the organelle, the Px I-III KO cells were studied by immunofluorescence microscopy. After 1 hr in Trolox-free medium, many cells still displayed a proper MitoTracker signal but surprisingly weak immune staining for the mitochondrial matrix 2-Cys-peroxiredoxin (mtTXNPx) (*Figure 2A*). After 2 hr in the absence of Trolox, the majority of cells had lost the MitoTracker signal and the mtTXNPx antibodies visualized some bright spots in addition to the faint mitochondrial staining. Respective observations were made when antibodies against two other matrix proteins, lipoamide dehydrogenase and acetate-succinate-CoA-transferase, were used (not shown). The reasons for the impaired and later punctuated staining of mitochondrial matrix proteins in the fixed and permeabilized cells are not clear. One may speculate that the antibody penetration was hampered and later, upon progressive condensation of the matrix, the proteins became concentrated within distinct areas. When antibodies against the voltage-dependent anion channel (VDAC) were used, this phenomenon was not observed. Instead, cells that after 1 hr in the absence of Trolox had already lost their MitoTracker Red signal, retained the VDAC staining (*Figure 2B*). Even after 2 hr, when the majority of cells no longer showed MitoTracker staining and had an overall swollen cell body, the antibodies against the outer mitochondrial membrane protein still visualized the tubular mitochondrion.

To get a deeper insight in the morphological changes, the Px I-III KO cells were subjected to transmission electron microscopy. Cells kept in the presence of Trolox or for only 30 min in Trolox-free medium were indistinguishable from wildtype parasites. These cells are characterized by a highly elongated cell body with densely packed cytoplasm and distinguishable subcellular structures such as the nucleus, mitochondrion, Golgi apparatus, ER, acidocalcisomes, glycosomes and lipid droplets. Staining of the mitochondrion and cristae was comparable or even lighter than that of the cytoplasm (*Figure 3A*). When the Px I-III KO cells were incubated for ≥1 hr in Trolox-free medium, cells appeared that still had an elongated shape, but, compared to the controls, lighter cytoplasm and a more electron-dense mitochondrion (*Figure 3B and C*, *Figure 3—figure supplement 1*). Both morphological changes appeared to be linked as cells with darkened mitochondrion but normal cytosol or, vice versa, with normal mitochondrion but lighter cytosol were hardly or not detectable (*Figure 3D*). Other organelles were unaffected (*Figure 3B*, *Figure 3—figure supplement 1*). The lighter cytosol is probably due to the gradual increase of the cell volume. Indeed, upon prolonged incubation in the absence of the antioxidant, light microscopy of the Px I-III KO cells revealed an increase of swollen cells (not shown). The dark mitochondria contained a growing number of less electron-dense area (*Figure 3C*). These white structures appeared within the organelle but were

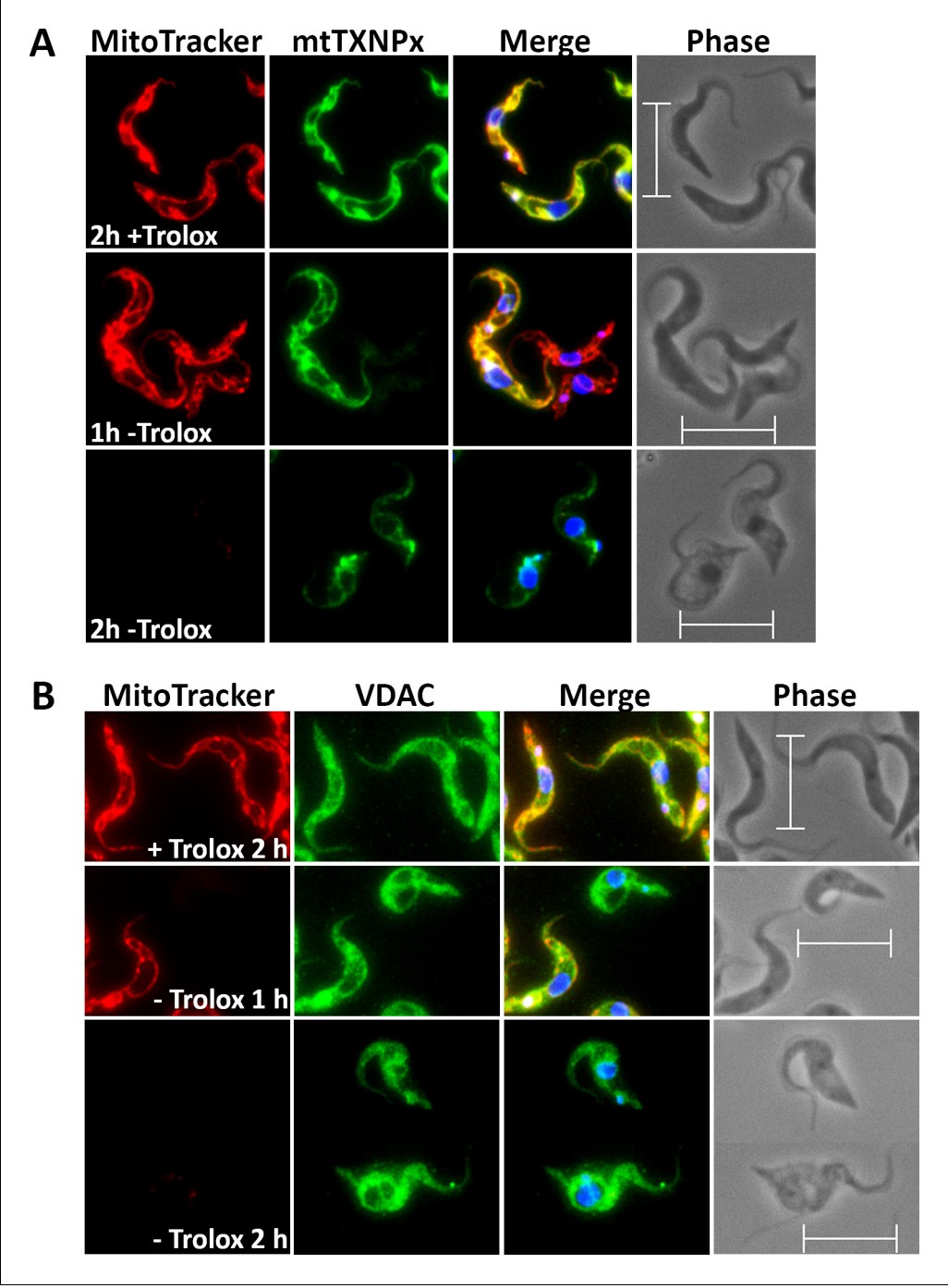

**Figure 2.** Px I-III KO cells are affected in mitochondrial matrix – but not outer membrane – immuno-staining. The parasites were incubated for 1 or 2 hr in medium ± Trolox, treated with MitoTracker (red) and subjected to immunofluorescence microscopy using antibodies against (**A**) mtTXNPx (green) or (**B**) VDAC (green). Nuclear and kinetoplast DNA were visualized by DAPI staining (blue). Merge, overlay of the respective three signals. Phase, phase contrast image. Scale bare 10 µm.

DOI: https://doi.org/10.7554/eLife.37503.005

The following figure supplement is available for figure 2:

**Figure supplement 1.** Mitochondrial damage precedes plasma membrane leakage.

DOI: https://doi.org/10.7554/eLife.37503.006

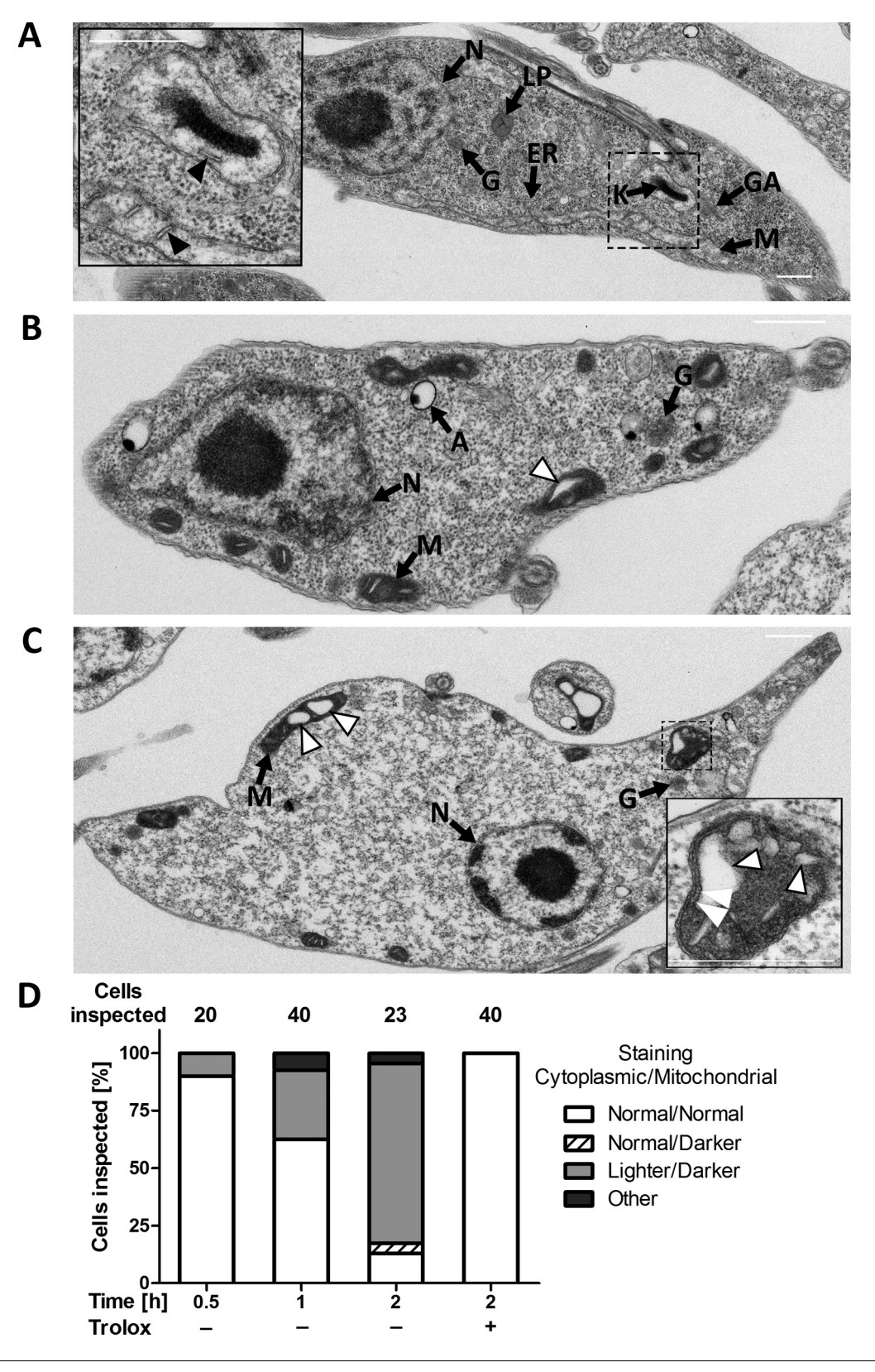

**Figure 3.** PC Px I-III KO *T. brucei* undergo strinking ultrastructural changes at their single mitochondrion. Parasites kept in the presence or absence of Trolox were fixed, processed and subjected to transmission electron
*Figure 3 continued on next page*

*Figure 3 continued*

microscopy as described under Materials and methods. Electron micrographs of representative cells incubated for (**A**) 2 hr in Trolox-containing medium and (**B and C**) 2 hr in Trolox-free medium. N, nucleus; LP, lipid droplet; G, glycosome; ER, endoplasmic reticulum; K, kinetoplast; GA, Golgi apparatus; M, mitochondrion; A, acidocalcisome; black arrow heads, normal cristae; white arrow heads, enlarged cristae. The inserts in (**A**) and (**C**) show higher magnifications of mitochondria to highlight the normal and altered morphology, respectively. The double white arrow heads point to the three membranes that surround a dilated cristae. Scale bars 500 nm. (**D**) Quantification of different phenotypes observed. In the presence of Trolox or for ≤0.5 hr without the antioxidant, the mutant cells were indistinguishable from wild type parasites (not shown). The cytosol was densely packed and the mitochondrion had a comparable or even lower electron density, defined as normal. In the absence of Trolox, the Px I-III KO parasites displayed a time-dependent darkening of the mitochondrion and lightening of the cytosol. Only parasites that clearly displayed an elongated cell body in the electron micrographs were incorporated in the analysis. The number of cells inspected at the different time points is given above the columns. The percentage of each phenotype in the total number of inspected cells is depicted.

DOI: https://doi.org/10.7554/eLife.37503.007

The following figure supplement is available for figure 3:

**Figure supplement 1.** Short-term incubation of Px I-III KO cells in Trolox-free medium results in mitochondrial alterations without changes at other organelles, whereas prolonged incubation can finally lead to plasma membrane blebs.

DOI: https://doi.org/10.7554/eLife.37503.008

never found to protrude out into the cytosol. Notably, high magnification revealed three membranes that surrounded the white bleb (*Figure 3C*, insert). This strongly suggested that the outer mitochondrial membrane remained intact and the blebs were dilated cristae. The phenotype was reminiscent of a condensed matrix and enlarged intermembrane space as observed in isolated mitochondria after transfer into high osmotic medium (*Cortese et al., 1991*). After 2 hr in Trolox-free medium, single cells were found that showed the release of vesicles from the plasma membrane (*Figure 3—figure supplement 1*).

## Px I-III-deficient cells generate mitochondrial oxidants and lose ATP

MitoSOX Red is widely used as indicator for mitochondrial superoxide production. Oxidation of the compound and binding of the oxidation products to nucleic acids is associated with strong red fluorescence. To assess the cellular localization of the sensor, Px I-III KO cells were treated with MitoSOX in Trolox-supplemented medium, incubated in medium ± Trolox, stained with MitoTracker Green and Hoechst 33342 and subjected to fluorescence microscopy. Cells in Trolox-supplemented medium showed a very small single red dot which co-localized with the DAPI signal for the kinetoplast and thus binding of some probably photo-oxidized sensor to the mitochondrial DNA (*Figure 4A*, upper panel). When kept for 2 hr in Trolox-free medium, most of the MitoSOX-treated Px I-III KO cells had still a normal morphology but many of them displayed a more intense kinetoplast fluorescence (*Figure 4A*, lower panels). A nuclear staining was not observed. This strongly suggests that at the beginning of the lethal process MitoSOX senses oxidants that are generated within the mitochondrial matrix. Swollen cells that appear upon prolonged incubation without protecting agents revealed an overall week red fluorescence and intense staining of the kinetoplast or nucleus or both structures. Evidently, when the mitochondrial membrane potential is abrogated, MitoSOX leaks out and loses its specificity for the mitochondrial matrix.

As proof of principle, Px I-III KO cells in Trolox-supplemented medium were loaded for 10 min with MitoSOX, treated with 2 µM antimycin and subjected to flow cytometry. In agreement with data published for other cells (*Mukhopadhyay et al., 2007*), the treatment resulted in a 5-fold fluorescence increase (not shown). The Px I-III KO cells were then treated ± MitoSOX in Trolox-supplemented medium, transferred into medium ± Trolox, incubated for up to 5 hr and treated with DAPI. When kept in the presence of Trolox, the MitoSOX-loaded cells displayed a basal fluorescence comparable to the auto-fluorescence of entirely untreated cells (-Trolox, - MitoSOX). Incubation in Trolox-free medium resulted in increased red fluorescence (*Figure 4B*, left). After 3 or 5 hr, the cells displayed a 3-fold higher MitoSOX fluorescence than the Trolox-supplemented control (*Figure 4C*).

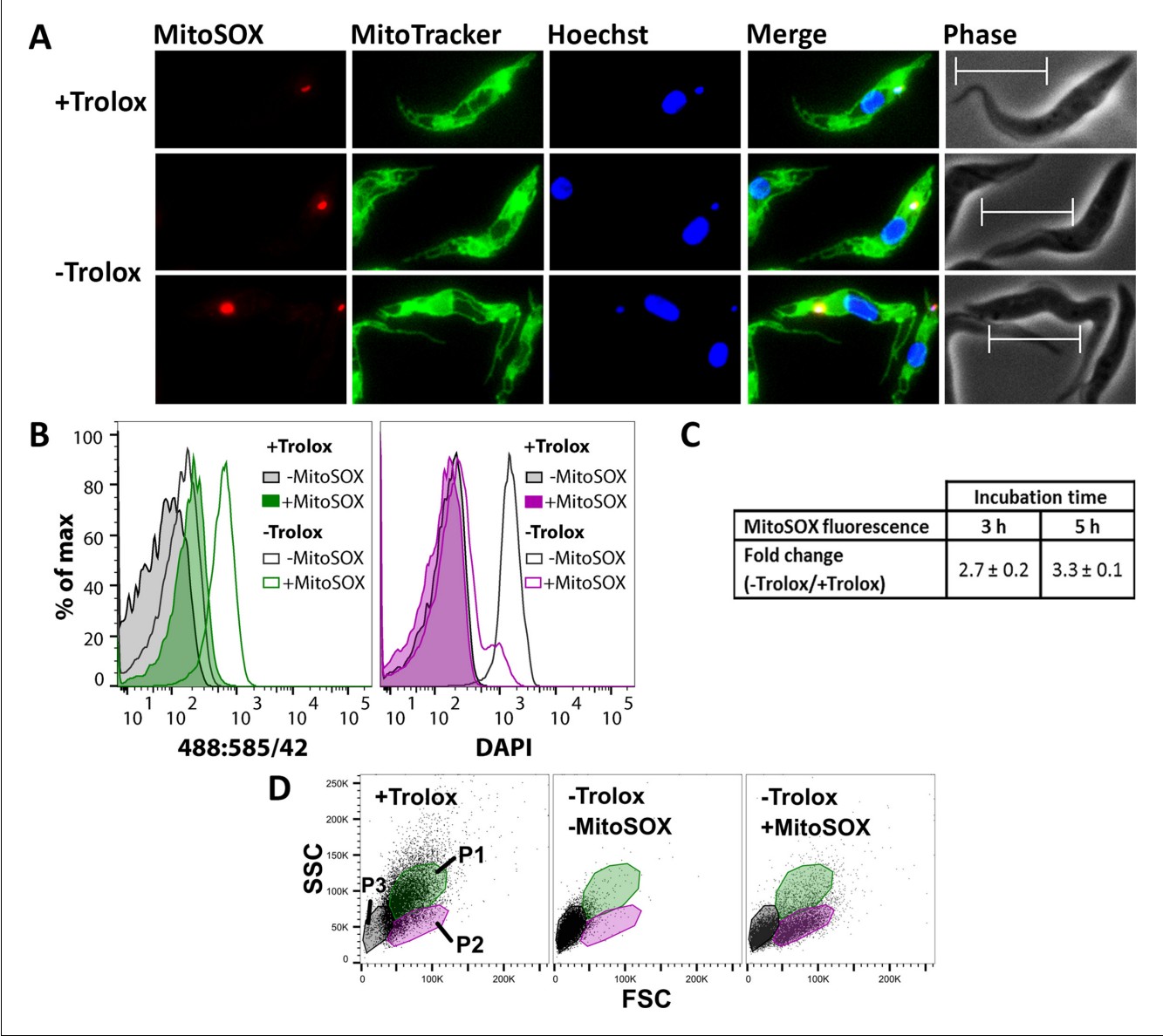

**Figure 4.** MitoSOX Red exerts a dual effect on the Px I-III KO cells, sensing oxidant production and acting as protecting agent. (**A**) The cells were pre-loaded with MitoSOX in Trolox-containing medium, transferred into medium ± Trolox, incubated for 2 hr, and stained with MitoTracker Green. Nuclear (large dot) and kinetoplast (small dot) DNA were visualized by Hoechst 33342 staining. Representative fluorescence microscopy images are depicted. Merge, overlay of all three signals. Phase, phase contrast image. Cells kept in the presence of Trolox displayed a very small red MitoSOX signal that co-localized with the kinetoplast (upper panel). After 2 hr in the absence of Trolox, most cells still had normal morphology but many of them displayed a more intense kinetoplast MitoSOX signal (lower two panels). Scale bar 10 μm. (**B–D**) The cells were incubated ± MitoSOX in Trolox-containing medium, transferred into medium ± Trolox, incubated for up to 5 hr, stained with DAPI, and subjected to flow cytometry. (**B**) Representative histograms of the 488:585/42 (ex:em) channel (MitoSOX signal of the treated cells and auto-fluorescence of non-treated cells) and DAPI signal of all single cells from samples kept for 5 hr ± Trolox, with or without loading with MitoSOX. (**C**) MitoSOX fluorescence change between cells kept in the absence and presence of Trolox. The data represent the mean ± error of the mean of two independent experiments. (**D**) Representative FSC:SSC DotPlots of cells kept for 5 hr in the presence or absence of Trolox, with or without loading with MitoSOX. The cells were gated in three sub-populations (P1-3) representing cells with normal FSC and SSC (P1; green), those with reduced SSC (P2; magenta), and severely damaged or dead cells (P3; grey).
DOI: https://doi.org/10.7554/eLife.37503.009

The following figure supplement is available for figure 4:

**Figure supplement 1.** The P2 sub-population displays the strongest increase in MitoSOX fluorescence.
DOI: https://doi.org/10.7554/eLife.37503.010

Strikingly, after 5 hr in the absence of Trolox but presence of MitoSOX, just a minute fraction of cells showed increased DAPI fluorescence. Only when incubated in the absence of both Trolox and MitoSOX, the cells incorporated DAPI (*Figure 4B*, right). To further evaluate the putative protecting effect of MitoSOX, light scattering of the Px I-III KO cells was followed. As expected, cells kept in the presence of Trolox had normal FSC and SSC (P1 population) whereas those in standard medium showed low FSC and SSC (P3) due to severe damage. When pre-loaded with MitoSOX, indeed, a considerable portion of the Px I-III KO cells in Trolox-free medium remained in the P1 or P2 fractions even after 5 hr of incubation (*Figure 4D*).

Next, we wanted to dissect which cell fraction was mainly responsible for the oxidant production. After 5 hr in Trolox-free medium, all three sub-populations had increased MitoSOX fluorescence but only the P3 fraction showed high DAPI fluorescence in accordance with the presence of severely damaged or dead cells (*Figure 4—figure supplement 1*). The strongest increase in MitoSOX fluorescence was observed for the P2 population which suggests that oxidant production was highest in cells with impaired motility and/or morphological alterations but still intact plasma membrane (DAPI-negative). The data indicate that in the peroxidase-deficient parasites production of soluble oxidants starts within the mitochondrial matrix and then rapidly spreads all over the cell and MitoSOX acted as both oxidant sensor and protecting antioxidant.

Finally, putative changes in the cellular ATP level were measured. The first analysis was carried out in SDM-79 medium. This could be the reason why lysis of the Px I-III KO cells in the absence of a protecting agent was slightly delayed when compared to studies in MEM-Pros medium. After 2 hr in the absence of Trolox, the ATP level had dropped by more than 50%, but virtually all cells retained normal morphology (*Figure 5A*). When only highly motile cells were considered, the loss in ATP essentially mirrored the decline in the cell number. Since flagellar movement requires high levels of ATP, motility is expected to be rapidly affected upon ATP depletion (*Langousis and Hill, 2014*). Thus, a substantial fraction of parasites ceased to move but retained a virtually normal shape even though the ATP level was already significantly decreased. On the other hand, a small number of cells appeared to be swollen but still had a beating flagellum. This may be due to the degree to which the ATP level drops in different areas of the highly elongated cell. In an individual parasite, ATP depletion may primarily affect flagellum beating, which originates at its tip, or the ion pumps in the plasma membrane. To rule out any bias upon cell counting, the experiment was repeated using MEM-Pros medium and PI staining to follow cell viability. After 1 hr in Trolox-free medium, the

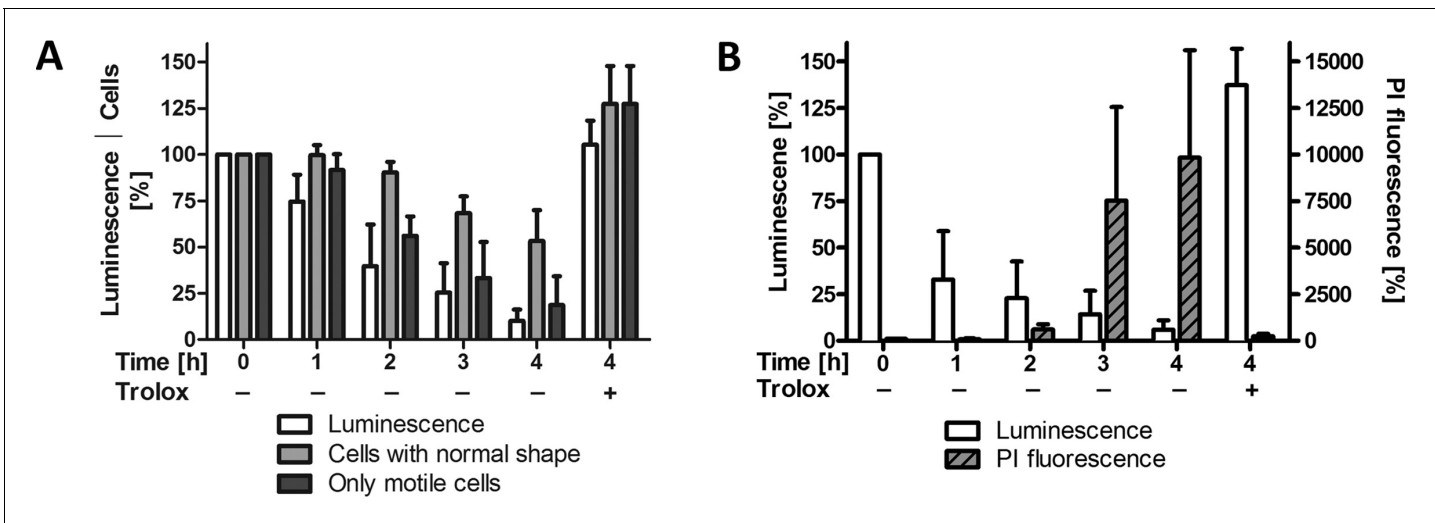

**Figure 5.** Cellular ATP levels rapidly decrease when Px I-III KO cells are in medium lacking a protecting agent. The cells were incubated ± Trolox in (**A**) SDM-79 medium or (**B**) MEM-Pros medium. After 0 to 4 hr, aliquots of each sample were removed and (**A**) cells with normal morphology or only highly motile cells were counted, or (**B**) the cells were treated with PI and the fluorescence was measured by flow cytometry. The remaining cells were treated with ATPlite one-step solution and the luminescence was measured in the plate reader. The data are given as percentage of the respective value at 0 hr that was set as 100%. They represent the mean ± SD of three independent experiments.
DOI: https://doi.org/10.7554/eLife.37503.011

cellular ATP had dropped by 50% but no PI staining was observed (**Figure 5B**). After 3 hr, the cells incorporated PI. At this time point the remaining ATP level was diminished to 20% of the control. Taken together, in the absence of a protecting agent, PC Px I-III KO cells rapidly lose ATP. This correlates with an impaired motility and altered morphology of the cells but not an immediate plasma membrane leakage.

## Mitochondrial membrane peroxidation is an early event in the lethal phenotype of the Px I-III-deficient cells

As shown in **Figure 1**, in the absence of a lipophilic antioxidant or iron chelator, Px I-III KO cells undergo lipid peroxidation. This analysis did, however, not allow to identify the subcellular membrane primarily affected. MitoPerOx is a derivative of BODIPY 581/591 C11 that is taken up into mitochondria of living cells where it is targeted to the interior surface of the inner membrane (**Prime et al., 2012**). To evaluate the subcellular localization of the sensor, Px I-III KO cells in Trolox-supplemented medium were treated with MitoPerOx and stained with MitoTracker Green. Life cell fluorescence microscopy revealed a perfect overlay of both signals confirming a strong enrichment

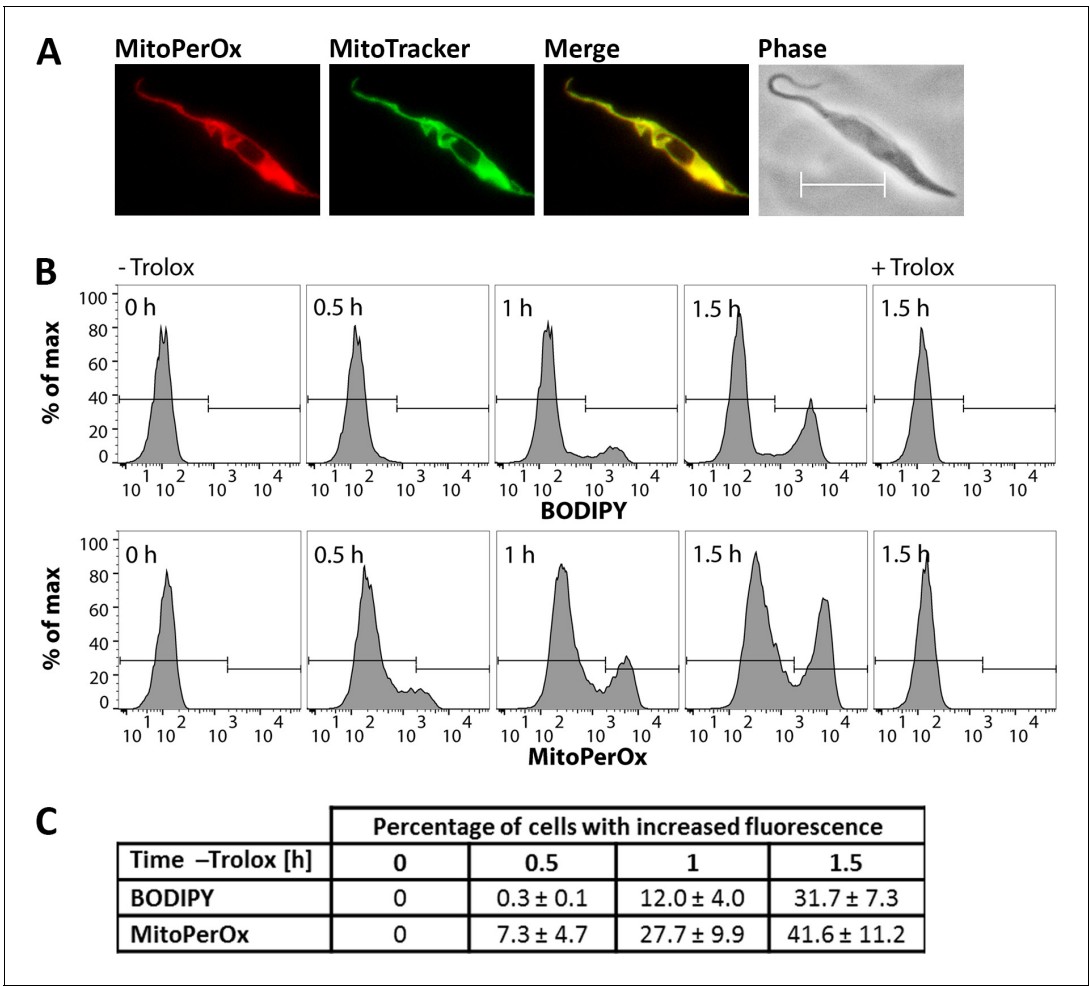

**Figure 6.** Lipid peroxidation in the Px I-III KO cells originates at the mitochondrion. (A) Px I-III KO parasites in Trolox-supplemented medium were incubated for 1.5 hr with MitoPerOx, stained with MitoTracker Green and subjected to life cell fluorescence microscopy. Representative images are depicted. Merge, overlay of both signals. Phase, phase contrast image. Scale bar 10 µm. (B) The cells were transferred into medium ± Trolox. BODIPY or MitoPerOx was added and after different times cells were analyzed by flow cytometry. Histograms of BODIPY (upper panel) and MitoPerOx (lower panel) fluorescence of all single cells at 520 nm from a representative experiment are depicted. (C) Quantitative analysis of the percentage of cells with increased BODIPY and MitoPerOx signals (gating as depicted in B). The data represent the mean ± SD of three independent experiments.
DOI: https://doi.org/10.7554/eLife.37503.012

of MitoPerOx in the mitochondrion (*Figure 6A*). The cells were then incubated in medium ± Trolox that was supplemented with either BODIPY or MitoPerOx. Both sensors revealed a time-dependent fluorescence increase in accordance with lipid peroxidation (*Figure 6B*). In the MitoPerOx-treated sample, after 30 min in Trolox-free medium, cells with increased fluorescence appeared whereas in the BODIPY-treated cells, the fluorescence started to increase after 60 min (*Figure 6C*). Differences in the sensitivity of the probes can be ruled out. The use of 2 μM BODIPY and 100 nM MitoPerOx ensured a very similar overall fluorescence. Thus, the mitochondria-targeted probe was slightly faster oxidized than the untargeted sensor reporting on all cellular membranes which suggests that lipid peroxidation starts within the matrix-facing leaflet of the inner mitochondrial membrane.

## Overexpression of a mitochondrial superoxide dismutase attenuates the cell death program of Px I-III-deficient parasites

As shown in *Figure 4*, the PC Px I-III KO cells generate mitochondrial oxidants but the precise nature of the products is not known. The red fluorescence measured could arise from reaction of MitoSOX with superoxide but also other oxidants such as iron/$H_2O_2$ (*Kalyanaraman et al., 2012*). African try-panosomes express four iron-superoxide dismutases of which two (SODA and SODC) are mitochondrial proteins. BS *T. brucei* in which the mRNA of SODA is down-regulated exhibit increased sensitivity towards the superoxide-inducing agent paraquat (*Wilkinson et al., 2006*).

The coding region of SODA was cloned into a vector that allows the tetracycline (Tet)-inducible expression of the protein with C-terminal myc$_2$-tag. PC Px I-III KO cells were transfected with the pHD1700/*sodA-c-myc$_2$* construct and five cell lines obtained by serial dilutions (for details see Materials and methods). Western blot analysis revealed for the induced Px I-III KO/SODA-myc cells expression of SODA-myc but, to a low level, also in the non-induced cells, indicating leaky expression (*Figure 7A*). Since antibodies against the authentic *T. brucei* SODA were not available and those against the *T. cruzi* ortholog did not detect the protein in total lysates of *T. brucei*, the cellular level of overexpression could not be assessed. Immunofluorescence microscopy of the Px I-III KO/SODA-myc cells with myc antibodies revealed a perfect overlay with the MitoTracker Red signal (*Figure 7B*). In comparison, antibodies against cytosolic peroxiredoxin evenly stained the whole cell body. The apparent partial co-localization of SODA-myc with the cytosolic marker is explained by the fact that the mitochondrion occupies 25% of the total cell volume (*Böhringer and Hecker, 1975*).

In medium lacking any protecting agent, the non-induced Px I-III KO/SODA-myc cells died within 4 hr as did the parental Px I-III KO cells. The level of SODA-myc in the non-induced cells was not sufficient for protection. The induced Px I-III KO/SODA-myc cells displayed a slightly delayed lysis (*Figure 7C*). In the presence of 25 μM Dfx, a concentration that protected the non-induced and parental cell lines only partially, the induced Px I-III KO/SODA-myc cells remained viable. To evaluate the effect of SODA-overexpression in more detail, induced and non-induced Px I-III KO/SODA-myc cells were incubated in medium ± 25 μM Dfx, treated with PI and subjected to flow cytometry. Overexpression of SODA or the presence of 25 μM Dfx partially prevented PI incorporation whereas in combination both treatments caused strong protection, in accordance with the cell counting results. Notably, in the absence and presence of Dfx, expression of SODA had a protecting effect (*Figure 7D*). Overexpression of the mitochondrial SOD rendered the peroxidase-deficient parasites less sensitive to an iron-induced cell death. Superoxide oxidizes [4Fe4S]-cluster proteins, a process that generates hydrogen peroxide and releases iron (*Winterbourn, 2008*). The ectopic expression of SODA may lower the concentration of free iron either by diminishing the damage of iron sulfur clusters or because of the overexpression of an iron-containing protein. In any case, the data indicate a crucial role of mitochondrial matrix iron in the cell death of the Px I-III-deficient parasites.

## Mitochondrial iron plays a crucial role in the cell death of Px I-III-deficient parasites

The involvement of iron is the primary characteristic of cells undergoing ferroptosis, however the subcellular site of iron that triggers the death program is not clear (*Doll and Conrad, 2017*). RPA is a mitochondria-targeted iron sensor. It is composed of a hydrophobic cationic rhodamine B moiety that mediates the uptake into mitochondria and a phenanthroline part that reacts with Fe (II). Quenching of the RPA fluorescence is a selective indicator of mitochondrial chelatable iron in intact

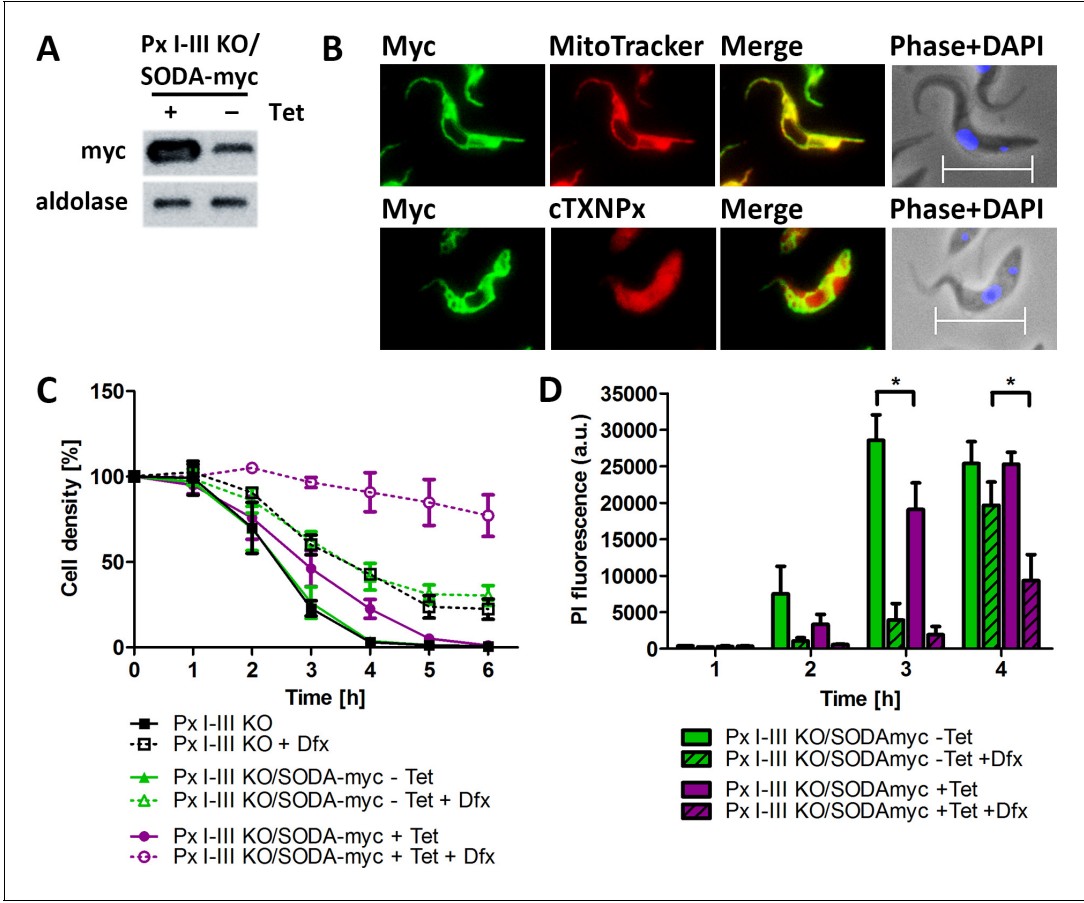

**Figure 7.** Px I-III KO cells that overexpress mitochondrial SODA require less Dfx for survival. (**A**) Total lysates of $5 \times 10^6$ Px I-III KO/SODA-myc cells cultured for 18 hr ± tetracycline (Tet) were subjected to Western blot analysis using antibodies against myc as well as aldolase for loading control. (**B**) Px I-III KO/SODA-myc cells cultured for 18 hr in the presence of Tet were treated with MitoTracker (red, upper panel) followed by antibodies against Myc (green) or simultaneously with antibodies against Myc and cytosolic 2-Cys-peroxiredoxin (cTXNPx) (red; lower panel). Nuclear and kinetoplast DNA were stained with DAPI (blue) and the cells were subjected to immunofluorescence microscopy. Merge, overlay of the Myc with the MitoTracker or cTXNPx signal. Phase + DAPI, phase contrast image with the nucleus (large dot) and the kinetoplast (small dot) visualized by DAPI staining. Scale bar 10 μm. (**C**) Px I-III KO and Px I-III KO/SODA-myc cells cultured for 18 hr ± Tet in Trolox-supplemented medium were transferred into Trolox-free medium and incubated ± Tet and ± 25 μM Dfx. Every hour viable cells were counted. The data represent the mean ± SD of three independent experiments. (**D**) Px I-III KO/SODA-myc cells cultured for 18 hr ± Tet in Trolox-supplemented medium were transferred into Trolox-free medium and incubated ± Tet and ± 25 μM Dfx for 1–4 hr starting with the longest time point. Cells were stained with PI and subjected to flow cytometry. Data show the mean + SEM of the PI fluorescence of all single cells from two experiments. Data were analyzed by two-way ANOVA with Bonferroni post-test, *p<0.05.
DOI: https://doi.org/10.7554/eLife.37503.013

cells (*Petrat et al., 2002*; *Rauen et al., 2007*). RPAC has the same fluorophore and linker but lacks iron-chelating capacity and can serve as control. Preliminary experiments revealed that RPAC-treated parasites had a much higher fluorescence than cells loaded with RPA. Therefore, in experiments following the fluorescence, the cells were treated with different concentrations of the fluorophores. Px I-III KO cells were loaded with 150 nM RPA or 10 nM RPAC in PBS, followed by 15 min incubation in Trolox-containing medium to ensure the selective uptake of the sensor into the mitochondrion. Afterwards, the cells were stained with MitoTracker Green and subjected to fluorescence microscopy. The RPA and RPAC fluorescence coincided with the MitoTracker signal confirming a mitochondrial localization (*Figure 8A*). Next, the cells were loaded with 150 nM RPA or 10 nM RPAC and then incubated in medium ± Trolox (*Figure 8B*). When the cells were kept in the presence of Trolox, both sensors displayed clear mitochondrial localization. After 2 hr in the absence of Trolox, the RPA-treated cells were virtually unaffected (not shown) whereas the RPAC-treated sample displayed swollen cells with comparably faint and unspecific fluorescence. When kept for 3 hr in Trolox-

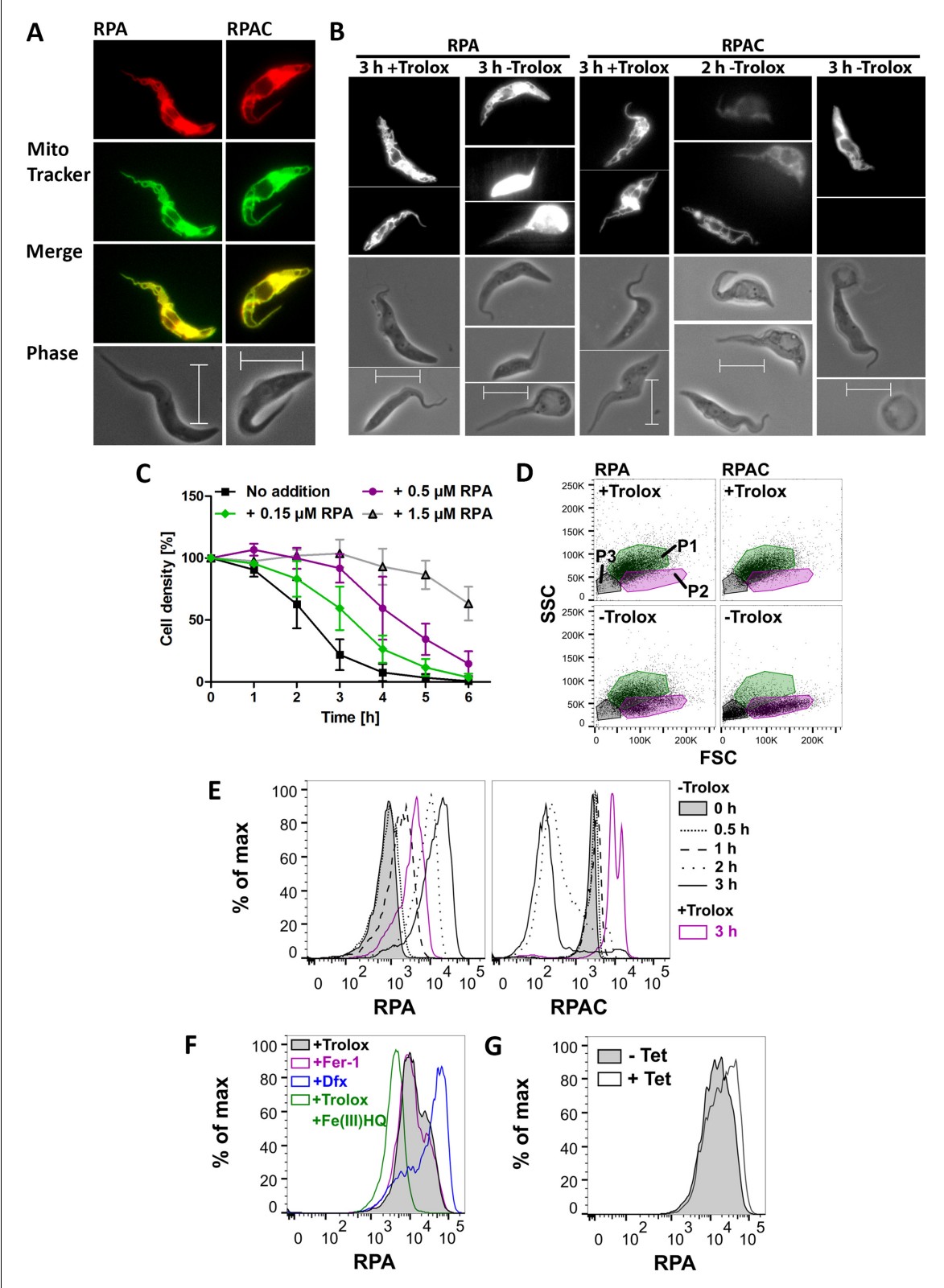

**Figure 8.** Mitochondrial iron is involved in the death phenotype of Px I-III-deficient cells. (**A**) Cells were pre-loaded with 150 nM RPA or 10 nM RPAC in PBS, re-transferred into Trolox-supplemented medium, stained with MitoTracker Green and subjected to life cell imaging. Merge, overlay of the RPA and RPAC fluorescence, respectively, with the MitoTracker signal. Phase, phase contrast image. (**B**) Cells pre-loaded with 150 nM RPA or 10 nM RPAC were transferred into medium ± Trolox and incubated for up to 3 hr. The upper panels show the RPA or RPAC fluorescence, the lower ones the

*Figure 8 continued on next page*

*Figure 8 continued*

respective phase contrast pictures. (**A and B**) Representative parasites are depicted. Scale bars 10 µm. (**C**) The cells were pre-loaded with 0.15, 0.5 or 1.5 µM RPA, transferred into medium – Trolox and incubated for 6 hr. Every hour, viable cells were counted. Cells not treated with RPA (no addition) served as control. The data are the mean ± SD of three independent experiments. (**D**) FSC:SSC dot plots of cells that were treated with 150 nM RPA or RPAC and incubated for 3 hr ± Trolox. P1 (green), P2 (magenta) and P3 (grey) gates determined as defined in *Figure 4*. A representative histogram of two independent experiments is shown. (**E**) RPA and RPAC fluorescence of cells stained with 50 nM RPA or 1 nM RPAC and incubated ± Trolox for 0–3 hr. (**F**) RPA fluorescence of cells incubated for 2 hr in medium supplemented with 100 µM Trolox ± 100 µM Fe(III)/HQ complex, 100 nM ferrostatin-1 or 100 µM Dfx and stained with 50 nM RPA. (**G**) RPA fluorescence of Px I-III KO/SODA-myc cells grown for 20 hr in the presence or absence of Tet and stained with 50 nM RPA. (**E–G**) Representative histograms from one of at least three independent experiments are depicted.

DOI: https://doi.org/10.7554/eLife.37503.014

free medium, most of the RPA-loaded cells appeared swollen and highly fluorescent while the majority of RPAC-treated parasites were rounded up and had lost their fluorescence. This time-dependent increase of RPA fluorescence and decline of RPAC fluorescence probably reflects leakage of the sensors to the extra-mitochondrial space caused by the loss of the mitochondrial membrane potential.

The observation that after 3 hr in Trolox-free medium most of the RPA-loaded cells were only swollen whereas the RPAC-loaded control cells were highly damaged suggested a putative protecting effect by the iron-chelating sensor. To analyze this in more detail, the Px I-III KO cells were pre-loaded with different concentrations of RPA, transferred into Trolox-free medium and cell viability was followed for 6 hr by counting living cells. The presence of 150 nM RPA slowed down cell lysis; and 1.5 µM RPA exerted a protecting effect comparable to that observed with 50 µM Dfx (*Figure 8C* and *Figure 1—figure supplement 1*). The parasites were then stained with 150 nM RPA or RPAC, incubated for 3 hr ± Trolox and inspected by light scatter analysis. Indeed, after loading with RPA, but not with RPAC, a large portion of cells retained normal FSC and SSC (*Figure 8D*) indicating that chelation of mitochondrial iron by RPA protected the cells.

Next the cells were loaded with 50 nM RPA or 1 nM RPAC and then incubated for different times in medium ± Trolox and subjected to flow cytometry (*Figure 8E*). At this concentration, RPA exerted an only minor protecting effect. When the Px I-III KO cells were kept for ≥2 hr in the absence of protecting agent, the RPA fluorescence strongly increased and the RPAC signal decreased. That both 3 hr + Trolox probes had a higher fluorescence compared to the respective 0 hr – Trolox time points is probably a technical artifact as the 3 hr samples were treated with the dyes immediately after thawing the fluorescent compounds whereas the other samples were loaded later depending on the respective incubation times. Clearly, when comparing the fluorescence of the 3 hr ± Trolox samples, the RPA fluorescence was shifted to higher and the RPAC signal to lower values. Taken together, the fluorescence imaging and flow cytometry data indicate that upon loss of the mitochondrial membrane potential, the sensors leak out into the cytosol where the RPA fluorescence is de-quenched due to the lower iron concentration compared to the mitochondrion and that of RPAC is low due to dilution or loss.

To determine the fluorescence of the maximally quenched and de-quenched probe, respectively, Px I-III KO cells were pre-incubated in medium with Trolox ± Fe(III)/HQ complex, Fer-1 or Dfx, washed and loaded with 50 nM RPA. In the presence of Trolox or Fer-1, the cells displayed the same low fluorescence suggesting that the sensor was largely quenched by the mitochondrial chelatable iron (*Figure 8F*). When the cells were pre-treated with Fe(III)/HQ, a highly membrane-permeable complex that intracellularly is rapidly reduced (*Rauen et al., 2007*), the RPA fluorescence was shifted to even lower values which probably represents the maximally quenched signal. When instead of the lipophilic antioxidants, Dfx was used as protecting agent, the RPA fluorescence was strongly increased. This indicates that Dfx causes a decline of the mitochondrial iron available for reaction with RPA. Hoyes and Porter showed a time-dependent strong accumulation of radiolabeled Dfx in the soluble/cytosol fraction of the cell (*Hoyes and Porter, 1993*). This is further supported by our own data that in contrast to free Dfx, starch-coupled Dfx, which is restricted to the endosomal/lysosomal compartments (*Zhang and Lemasters, 2013*), is unable to protect the PC Px I-III KO cells (*Schaffroth et al., 2016*).

Finally, we studied if overexpression of mitochondrial SODA affects the mitochondrial iron level. Px I-III KO/SODA-myc cells, that were cultured for 20 hr in the presence or absence of Tet, were loaded with 50 nM RPA. As shown in *Figure 8G*, the fluorescence of the induced cells was slightly

higher when compared to the non-induced cells. The only minor shift is probably due to the fact that the cultures are not homogenous but contain cells that express SODA to various levels (as seen in the immunofluorescence analysis). Even so, the reproducibly observed shift to higher fluorescence suggests that overexpression of the iron-protein in the mitochondrial matrix lowers the iron available for chelation by RPA. Taken together, the data show that mitochondrial iron plays a crucial role in the death program of the peroxidase-deficient cells.

## Discussion

Here we report that trypanosomes that lack the lipid hydroperoxide-detoxifying tryparedoxin peroxidases undergo a cell death which is suppressed by iron chelation (Dfx) or lipophilic antioxidants (Trolox, α-Toc, Lpx-1, Fer-1) and involves accumulation of lipid hydroperoxides and thus fulfils all criteria defining ferroptosis (*Stockwell et al., 2017*; *Galluzzi et al., 2018*). The first alterations observed after transfer of Px I-III-deficient PC parasites into medium without protecting agent were mitochondrial lipid peroxidation, loss of the mitochondrial membrane potential, drop of cellular ATP and production of mitochondrial oxidants indicating that in these cells the ferroptosis-type death started at the mitochondrion. The implication of mitochondria in ferroptosis is still highly controversial and debated in the mammalian system. Human cancer cells undergoing erastin-induced ferroptosis retain normal ATP levels; and cells that lack a functional electron transport chain or are depleted of mitochondria can experience ferroptosis (*Dixon et al., 2012*; *Gaschler et al., 2018*). Clearly, an active mitochondrial electron transport chain is not a prerequisite for a cell to undergo ferroptosis. Yet, even in these cells, morphological changes such as the formation of smaller mitochondria with increased membrane density were observed (*Dixon et al., 2012*).

GPx4-deficient mouse embryonic fibroblasts (Pfa1 cells) show a breakdown of the mitochondrial membrane potential (*Seiler et al., 2008*), but initial lipid peroxidation occurs outside the mitochondrial matrix and involves a disruption of the outer mitochondrial membrane (*Friedmann Angeli et al., 2014*). In redox phospholipidomics, GPx4-inactivated Pfa 1 cells revealed increased levels of oxygenated derivatives for all major classes of phospholipids, with the exception of cardiolipin and lipid peroxidation was found to take place predominantly in endoplasmic reticulum (ER)-associated sites (*Kagan et al., 2017*). On the other hand, serum-induced ferroptosis in MEFs under amino acid starvation causes ATP depletion (*Gao et al., 2015*) and erastin-induced ferroptosis in neuronal cells is accompanied by a loss of the mitochondrial membrane potential and cellular ATP (*Neitemeier et al., 2017*). Strikingly, GPx4-deficient kidneys display a time-dependent formation of peroxidized cardiolipin (*Friedmann Angeli et al., 2014*) and mitochondria-targeted nitroxides are able to inhibit ferroptosis in a variety of tissue types and across multiple growth conditions demonstrating that mitochondrial lipid oxidation is critical in promoting the ferroptotic cell death (*Krainz et al., 2016*).

The mitochondrial phenotype of the peroxidase-deficient PC *T. brucei* is not a specialized property of these protozoa and their single mitochondrion. In the mammalian bloodstream form – which relies exclusively on glycolysis for energy supply and acquires iron by endocytosis of host transferrin – lack of the cytosolic peroxidases triggers an iron-dependent lipid peroxidation and cell death that originates at the lysosome (*Hiller et al., 2014*). A respective phenotype has been reported for human fibrosarcoma HT 1080 cells or lung cancer Calu-1 cells which are protected from erastin toxicity by inhibitors of lysosomal activities (*Torii et al., 2016*). Our results strongly suggest that ferroptosis can be initiated at distinct subcellular sites. The role of mitochondria in ferroptosis execution appears to depend on the individual cell type and even the culture conditions such as the use of high glucose medium. A crucial factor could be that rapidly proliferating cells such as many cancer and embryonic cells rely more on glycolysis whereas differentiated cells primarily use oxidative phosphorylation for ATP production. In addition, ferroptosis starting at one cellular compartment may rapidly spread to other sites. Lipid transport proteins disseminate oxidative stress between compartments by preferably trafficking peroxidized lipids to mitochondria (*Kriska et al., 2010*; *Vila et al., 2004*; *Friedmann Angeli et al., 2014*).

After loss of the mitochondrial membrane potential, Px I-III-deficient cells still displayed a tubular staining for VDAC, an integral protein of the outer mitochondrial membrane. The most prominent alterations revealed by electron microscopy were a condensation of the mitochondrial matrix and appearance of white blebs which probably reflect enlarged cristae. A rupture of the outer

membrane, described for mammalian cells after inactivation of GPx4 (*Friedmann Angeli et al., 2014*; *Doll et al., 2017*), was not observed. In accordance with the situation in ferroptotic mammalian cells, other subcellular structures, and in particular the nucleus, remained unaffected (*Stockwell et al., 2017*). In the absence of a protecting agent, the cytosol of the Px I-III-deficient parasites became progressively lighter due to swelling of the cell body which finally resulted in cell death. That mitochondrial membrane peroxidation is an early event in the death of the peroxidase-deficient cells is supported by the finding that MitoPerOx presented a faster oxidation kinetic than the untargeted BODIPY C11 sensor. The inner mitochondrial membrane is particularly susceptible to oxidative damage because of its very large surface area and the proximity to the superoxide-producing respiratory chain and the high content of peroxidation-sensitive unsaturated fatty acids in its phospholipids, notably cardiolipin (*Prime et al., 2012*). A comparative study on different phospholipids revealed cardiolipin with its two phosphate groups and thus most anionic phospholipid, as strongest binder of GPx4 (*Cozza et al., 2017*). *T. brucei* is rich in polyunsaturated fatty acids, with C22:4–6 and C20:2–5 as the most abundant species (*Richmond et al., 2010*) and also one of the cardiolipin species identified in isolated mitochondria contains a C22:6 fatty acid (*Guler et al., 2008*). This clearly creates a vulnerability of trypanosomes for membrane oxidation.

As a characteristic feature of cells prone to ferroptosis, the Px I-III-deficient cells retain full viability in the presence of an iron chelator but the subcellular site of the iron involved remained elusive. PC *T. brucei* lack a transferrin receptor but take up iron from ferric complexes via a 2-step mechanism in which ferric iron is reduced to ferrous iron and is subsequently transported (*Mach et al., 2013*). The parasites possess Mit1, a homolog of the iron transport protein mitoferrin-1 in the inner mitochondrial membrane of vertebrate cells. Mit1 is essential in PC *T. brucei* and depletion of its mRNA specifically affects the mitochondrion (*Mittra et al., 2016*). Here we present several lines of evidence that mitochondrial matrix iron triggers the cell death in the PC Px I-III-deficient trypanosomes. RPA, an iron-chelating sensor that is targeted to the mitochondrion, prolonged viability of the parasites. In Dfx-treated cells the sensor was de-quenched indicating that Dfx lowers the iron available for chelation by RPA. Radiolabeled Dfx was demonstrated not to be retained in the endosomal/lysomal compartments but to accumulate in the soluble/cytosolic fraction of mammalian cells (*Hoyes and Porter, 1993*). This is supported by our own data. Whereas free and starch-coupled Dfx equally protect BS *T. brucei* which lack the cytosolic peroxidases and show a lysosomal lethal phenotype, only free, but not starch-coupled Dfx, is able to protect the PC peroxidase-deficient cells (*Hiller et al., 2014*; *Schaffroth et al., 2016*). It is very likely that Dfx exerts its iron chelating and thus protecting effects in the cytosol. This could significantly alter our view on the mechanisms of ferroptosis that are partially based on the assumption that Dfx is membrane-impermeable and restricted to the lysosomal compartment (*Cao and Dixon, 2016*; *Gaschler et al., 2018*). Upon incubation of the Px I-III KO cells without any protecting agent, the RPA fluorescence became de-quenched, most probably because, due to breakdown of the mitochondrial membrane potential, the sensor leaks out into the cytosol where the chelatable iron concentration is expectedly lower than in the mitochondrial matrix. Further support for the role of matrix iron in the process came from the finding that overexpression of the mitochondrial iron-SODA lowered the Dfx concentration required to protect the peroxidase-deficient cells. Three putative protecting mechanisms could be envisaged: an attenuated generation of free iron from FeS clusters by superoxide, a diminished level of reactive iron by affecting the recycling of $Fe^{2+}$ from $Fe^{3+}$ and incorporation of iron into the newly expressed mitochondrial protein. Each of these mechanisms would lower the reactive iron level in the mitochondrial matrix. To our knowledge this is the first study in which the subcellular site of iron triggering ferroptosis has been identified.

Mechanisms that result in increased levels of poly-unsaturated fatty acids or cellular iron as well as those that decrease the capacity of the cell to detoxify peroxidized lipids such as depletion of GSH or GPx4 are able to trigger ferroptosis. Enzymatic effectors for example lipoxygenases can drive ferroptotic oxidation of poly-unsaturated fatty acids (*Seiler et al., 2008*; *Stockwell et al., 2017*). However, the lipoxygenase-mediated lethal effect is observed under GSH depletion conditions but not if ferroptosis is induced by GPx4 inactivation demonstrating that the presence of a lipoxygenase is not essential for the cell death program (*Yang et al., 2016*). As shown recently for diverse types of lipoxygenase inhibitors, there is a correlation between the anti-ferroptotic activity and reactivity as radical-trapping agents (*Shah et al., 2018*). Clearly, ferroptosis can be triggered spontaneously for instance by a rise in peroxidized lipids. There is a close link between the cellular

lipid composition and ferroptosis sensitivity. ACSL4 (acyl-CoA synthetase long-chain family member 4) is a critical determinant of the membrane lipid composition and sensitizes cells to ferroptosis whereas cells lacking ACSL4 show marked resistance to ferroptosis induction (*Doll et al., 2017*). The *T. brucei* genome encodes at least five acyl-CoA synthetases (ACSs). One of the four enzymes studied so far (ACS1) accepts arachidonic acid and 22:6 fatty acids as preferred substrates (*Jiang and Englund, 2001*) and the so far uncharacterized ACS5 (Tb927.10.3260) has been annotated as putative long-chain-fatty-acid-CoA-ligase. An interesting aspect of future studies will be the putative role of long-chain ASCs in triggering ferroptosis in these protozoa.

As shown here, ferroptosis is not restricted to mammalian cells (*Seiler et al., 2008*; *Dixon et al., 2012*; *Friedmann Angeli et al., 2014*; *Stockwell et al., 2017*) and plants (*Distéfano et al., 2017*). The regulated cell death occurs in trypanosomes, one of the earliest branching eukaryotes. The incorporation of polyunsaturated fatty acids into cell membranes was probably highly advantageous during evolution, allowing to modulate the membrane fluidity and functionality of membrane proteins and cells to adapt to different environments and temperatures. This is particularly important for African trypanosomes which during their digenetic life cycle switch between the 37°C bloodstream of a mammalian host and the 27°C midgut of the tsetse fly vector. The presence of polyunsaturated fatty acids in the membranes, however, requires sophisticated antioxidant systems. In most cells this is achieved by the GSH/GPx4 couple. Trypanosomes developed a defense system that is coupled to the unique trypanothione/tryparedoxin system and replaces the GSH/glutathione peroxidase GPx4 pair.

# Materials and methods

## Key resources table

| Reagent type (species) or resource | Designation | Source or reference | Identifiers | Additional information |
|---|---|---|---|---|
| Gene (*Trypanosoma brucei brucei*) | *sodA* | | TriTrypDatabase ID: Tb427.05.3350 | |
| Cell line (*Trypanosoma brucei brucei*) | WT PC and BS | PMID: 9108552 | WT PC and BS 449 Lister strain 427 | Culture-adapted *T. brucei* strains stably expressing the tetracycline repressor |
| Cell line (*Trypanosoma brucei brucei*) | PC Px I-III KO | PMID: 26374473 | | PC WT cells in which both alleles of the complete px locus are replaced by resistance casettes (blasticidin and puromycin) |
| Cell line (*Trypanosoma brucei brucei*) | PC Px I-III KO/SODA-myc | this work | | PC Px I-III KO cells that contain a Tet-inducible construct for SODA-c-myc$_2$ overexpression |
| Antibody | Rabbit anti-cTXNPx | PMID: 20826821 | | IF (1:1000) |
| Antibody | Guinea pig anti-mTXNPx | PMID: 29413965 | | IF (1:1000) |
| Antibody | Rabbit anti-aldolase | Christine Clayton, Heidelberg, Germany | | WB (1:20000) |
| Antibody | Rabbit anti-VDAC | André Schneider, Bern, Switzerland | | IF (1:500) |
| Antibody | Mouse anti-c-Myc (monoclonal) | Santa Cruz Biotechnology | sc-40, RRID:AB_627268 | WB, IF (1:200) |
| Antibody | HRP-conjugated goat anti-mouse IgGs | Santa Cruz Biotechnology | sc-2005, RRID:AB_631736 | WB (1:5000) |
| Antibody | HRP-conjugated goat anti-rabbit IgGs | Santa Cruz Biotechnology | sc-2004, RRID:AB_631746 | WB (1:10000) |
| Antibody | Alexa Fluor 488 goat anti-guinea pig IgGs | Thermo Fisher Scientific | A11073, RRID:AB_142018 | IF (1:1000) |

*Continued on next page*

*Continued*

| Reagent type (species) or resource | Designation | Source or reference | Identifiers | Additional information |
|---|---|---|---|---|
| Antibody | Alexa Fluor 488 goat anti-mouse IgGs | Thermo Fisher Scientific | A11001, RRID:AB_2534069 | IF (1:250) |
| Antibody | Alexa Fluor 488 goat anti-rabbit IgGs | Thermo Fisher Scientific | A11008, RRID:AB_143165 | IF (1:1000) |
| Antibody | Alexa Fluor 594 goat anti-rabbit IgGs | Thermo Fisher Scientific | A11012, RRID:AB_141359 | IF (1:1000) |
| Recombinant DNA reagent | pHD1700/grx2-c-myc2 (plasmid) | PMID: 20826822 | | |
| Recombinant DNA reagent | pHD1700/sodA-c-myc2 (plasmid) | this work | | |
| Sequence-based reagent | 5'CGATAAGCTTATG AGGTCTGTCATGATGC3' | this work | | Primer for amplification of *sodA* from genomic *T. brucei* DNA |
| Sequence-based reagent | 5'CGATGGATCCCTT CATAGCCTGTTCATAC3' | this work | | Primer for amplification of *sodA* from genomic *T. brucei* DNA |
| Peptide, recombinant protein | *T. brucei* trypanothione reductase (TR) | PMID: 24788386 | | |
| Peptide, recombinant protein | *T. brucei* tryparedoxin (Tpx) | PMID: 22275351 | | |
| Peptide, recombinant protein | *T. brucei* peroxidase (Px) | PMID: 18684708 | | |
| Chemical compound, drug | trypanothione and trypanothione disulfide | PMID:19477177 | | |
| Chemical compound, drug | Chlorhexidine diacetate Monohydrate | Fluka | 24800 | |
| Chemical compound, drug | (1S,3R)-RSL3 | José Pedro Friedmann Angeli and Marcus Conrad, Munich, Germany and Cayman Chemical | CAS: 1219810-16-8 | |
| Chemical compound, drug | Racemic mixture of RSL3 | José Pedro Friedmann Angeli and Marcus Conrad, Munich, Germany | | |
| Chemical compound, drug | Ferrostatin-1 | Sigma-Aldrich | SML0583 | |
| Chemical compound, drug | Liproxstatin-1 | Sigma-Aldrich | SML1414 | |
| Chemical compound, drug | Trolox | Sigma-Aldrich | 238813 | |
| Chemical compound, drug | Iron(III) chloride hexahydrate, $FeCl_3 \times 6H_2O$ | Merk | 31232 | |
| Chemical compound, drug | Deferoxamine mesylate | Sigma-Aldrich | D-9533 | |
| Chemical compound, drug | 8-Hydroxyquinoline | Sigma-Aldrich | 252565 | |
| Chemical compound, drug | (±)-α-Tocopherol | Sigma-Aldrich | T3251 | |

*Continued on next page*

*Continued*

| Reagent type (species) or resource | Designation | Source or reference | Identifiers | Additional information |
|---|---|---|---|---|
| Chemical compound, drug | ATPlite 1step solution | Perkin Elmer | 6016731 | |
| Chemical compound, drug | Hoechst 33342 | Walter Nickel, Heidelberg, Germany | | |
| Chemical compound, drug | 4',6-Diamidino-2-phenylindole (DAPI) | Sigma-Aldrich | D-8417 | |
| Chemical compound, drug | Rhodamine B-[(1,10-phenanthroline-5-yl)-aminocarbonyl]benzyl ester (RPA) | Squarix Biotechnology | ME043.1 | |
| Chemical compound, drug | Rhodamine B-[(phenanthren-9-yl)-aminocarbonyl]-benzylester (RPAC) | Squarix Biotechnology | ME046.1 | |
| Chemical compound, drug | BODIPY 581/591 C11 | Molecular Probes | D3861 | |
| Chemical compound, drug | MitoPerOx | Mike Murphy, Cambridge, UK | | |
| Chemical compound, drug | MitoTracker Red CMXRos | Molecular Probes | M7512 | |
| Chemical compound, drug | MitoTracker Green FM | Molecular Probes | M7514 | |
| Chemical compound, drug | MitoSOX Red | Molecular Probes | M36008 | |
| Chemical compound, drug | Propidium iodide (PI) | Molecular Probes | P3566 | |
| Chemical compound, drug | 2',7'-Dichlorodihydro fluorescein-diacetate (H2DCFDA) | Molecular Probes | D399 | |

## Materials

Tetracycline (Tet), Trolox, $\alpha$-tocopherol ($\alpha$-Toc), liproxstatin-1 (Lpx-1), ferrostatin-1 (Fer-1), deferox-amine mesylate (Dfx), DAPI, 8-hydroxyquinoline (HQ), iron (III) chloride x 6 $H_2O$, hemin, penicillin/streptomycin and phleomycin were purchased from Sigma, Munich, Germany. Hygromycin B was from Carl Roth, Karlsruhe, Germany. DCFH-DA and propidium iodide (PI) were purchased from Ther-moFisher, Schwerte, Germany. BODIPY 581/591 C11 (BODIPY), MitoSOX Red, MitoTracker CMXRos, and MitoTracker Green were from Life Technologies, Darmstadt, Germany. Rhodamine B-[(1, 10-phenanthroline-5-yl)-aminocarbonyl]benzyl ester (RPA) and rhodamine B-[(phenanthren-9-yl)-aminocarbonyl]-benzylester (RPAC) were from Squarix Biotechnology, Marl, Germany and fetal calf serum (FCS) from Biochrome, Berlin, Germany. A sample of Hoechst 33342 was kindly provided by Dr. Walter Nickel, Heidelberg, Germany. MitoPerOx was a kind gift from Dr. Mike Murphy, Cambridge, UK. Drs José Pedro Friedmann Angeli and Marcus Conrad, Munich, Germany, are kindly acknowledged for samples of (1S, 3R)-RSL3 and the racemic mixture of RSL3 (*Yang et al., 2014*). Additional (1S, 3R)-RSL3 was purchased from Cayman Chemical, Ann Arbor, Michigan.

Trypanothione and trypanothione disulfide (*Comini et al., 2009*) as well as tag-free recombinant *T. brucei* trypanothione reductase (TR) (*Persch et al., 2014*), tryparedoxin (Tpx) (*Fueller et al., 2012*) and peroxidase (Px) (*Melchers et al., 2008*) were prepared as described. Polyclonal rabbit antibodies against the cytosolic *T. brucei* 2-Cys-peroxiredoxins (cTXNPx) and guinea pig antibodies against the mitochondrial 2-Cys-peroxiredoxin (mtTXNPx) were obtained previously (*Ceylan et al., 2010*; *Ebersoll et al., 2018*). Polyclonal rabbit antibodies against *T. brucei* aldolase and VDAC and *T. cruzi* SODA were kindly provided by Drs Christine Clayton, Heidelberg, Germany, André Schneider, Bern, Switzerland, and Rafael Radi, Montevideo, Uruguay. Monoclonal mouse antibodies

against the c-Myc protein as well as horseradish peroxidase-conjugated goat antibodies against mouse and rabbit immunoglobulins G were purchased from Santa Cruz Biotechnology, Heidelberg, Germany.

## Cultivation and phenotypic analysis of different *T. brucei* strains

The parasites used in this work were all culture-adapted *Trypanosoma brucei brucei* of the cell line 449, descendants from the Lister strain 427 that stably express the tet repressor (*Cunningham and Vickerman, 1962*; *Biebinger et al., 1997*). The cells were kindly provided by Dr Christine Clayton, Heidelberg, Germany. BS cells were grown in HMI-9 medium at 37°C. If not stated otherwise, PC parasites were cultivated at 27°C in MEM-Pros medium as described previously (*Schlecker et al., 2005*). Distinct experiments were conducted in SDM79-CGGGPPTA medium (GE Healthcare, Munich, Germany). All media were supplemented with 10% heat-inactivated FCS, 50 units/ml penicillin, 50 µg/ml streptomycin and 0.2 µg/ml phleomycin. In addition, both media for PC parasites contained 7.5 µg/ml hemin but no glucose. PC cells lacking the complete Px I-III locus (Px I-III KO cells (*Schaffroth et al., 2016*) were routinely cultured in the presence of 100 µM Trolox (100 mM stock solution in ethanol). For phenotypic analyses, the parasites were harvested in the logarithmic growth phase, washed with PBS, resuspended in 5 ml medium adjusted to a density of about $5 \times 10^5$ cells/ml and subjected to different treatments. Stock solutions of Dfx (10 mM in PBS), Fer-1 (10 mM in DMSO), Lpx-1 (10 mM in DMSO), and α-Toc (10 mM in ethanol) were prepared and various concentrations added to the cells. After different times, viable cells (defined as those with normal, elongated shape) were counted in a Neubauer chamber. The mean and standard deviation (SD) of three independent experiments were calculated using GraphPad Prism software (GraphPad Software, La Jolla, CA).

## Cloning of Px I-III KO cells that overexpress mitochondrial SODA

The pHD1700 plasmid contains a hygromycin resistance gene and a cassette that allows for Tet-inducible expression of a protein with C-terminal myc$_2$-tag. The coding sequence of *sodA* without stop codon (714 bp) was amplified from genomic DNA of WT *T. brucei* by PCR using the primers 5'CGAT<u>AAGCTT</u>ATGAGGTCTGTCATGATGC3' and 5'CGAT<u>GGATCC</u>CTTCATAGCCTGTTCATAC3' (restriction sites underlined). The amplicon as well as plasmid pHD1700/*grx2-c-myc$_2$* (*Ceylan et al., 2010*) were digested with HindIII and BamHI, purified and ligated yielding pHD1700/*sodA-c-myc$_2$*. PC Px I-III KO cells were transfected with the NotI-linearized plasmid. After 24 hr cultivation, hygromycin was added and resistant cells were selected by serial dilutions as described in detail (*Musunda et al., 2015*).

## EC$_{50}$-Determination of RSL3 towards bloodstream *T. brucei*

The plate reader-based assay was carried out as described previously (*Fueller et al., 2012*; *Latorre et al., 2016*). Stock solutions of 10 mM (1*S*, 3*R*)-RSL3 and RSL3 racemate were prepared in DMSO. Each well contained 90 µl HMI-9 medium with 2500 cells/ml. The compounds were serially 1:3 diluted in 10 steps and 10 µl of the dilutions added to the cells to give final concentrations between 30 µM and 1 nM. 10% DMSO served as positive control. The highest DMSO concentration added with the compounds was 0.3% (which does not affect parasite viability, negative control). Since the Px-type enzymes are dispensable if the cells are kept in the presence of Trolox (*Hiller et al., 2014*), a second analysis was done in medium that was supplemented with 100 µM Trolox, 100 nM Fer-1 or 200 nM Lpx-1. After 24 hr, 48 hr and 72 hr cultivation, 50 µl ATPlite one step solution (PerkinElmer, Rodgau, Germany) was added and the luminescence measured in a Victor X4 plate reader (PerkinElmer) at room temperature. The luminescence intensities were plotted against the logarithmic compound concentrations and EC$_{50}$-values calculated using GraphPad Prism. Chlorhexidine (37 µM to 11 nM), a trypanocidal compound and known TR inhibitor (*Meiering et al., 2005*; *Beig et al., 2015*), served as positive control.

## In vitro kinetic analysis of RSL3

A putative irreversible inhibition of TR was studied essentially as described previously (*Otero et al., 2006*). In a total volume of 1 ml of 40 mM Hepes, 1 mM EDTA, pH 7.5, 1 µM *T. brucei* TR was incubated at 25°C with 100 µM RSL3 racemate in the presence of 500 µM NADPH. After different times

(0–120 min), 5 µl of the reaction mixture was removed, and the remaining activity measured in a 1 ml standard TR assay. Because of the dilution, reversible inhibition is not recorded under these conditions. Controls contained either buffer, TR and NADPH or buffer, TR and inhibitor in the pre-incubation mixtures.

The effect of RSL3 on the parasite peroxidase system was measured as described previously (*Fueller et al., 2012*). Shortly, in a total volume of 200 µl of 100 mM Tris, 5 mM EDTA, pH 7.6, 150 µM NADPH, 200 mU TR, 100 µM $T(SH)_2$, 10 µM Tpx, 60 nM Px and 40 µM RSL3 or 5 µl DMSO (control) were incubated at 25°C. After 1, 15, and 40 min, the reaction was started by adding 100 µM $H_2O_2$ and the absorption decrease followed at 340 nm. In a second approach, the pre-incubation mixture contained all components except the peroxidase, and the assay was started by adding both Px and $H_2O_2$.

## Flow cytometry

All treatments were performed at 27°C in the dark. Logarithmically growing Px I-III KO cells were harvested, split into samples of about $10^6$ cells, washed with cold PBS and re-suspended in 1 ml medium. For the detection of general cellular ROS, the cells were re-suspended in SDM-79 medium ± Trolox, incubated for 2 hr, washed and stained for 30 min with 10 µM $H_2DCFDA$ (10 mM stock solution in DMSO) in medium + Trolox. All following analyses were conducted in MEM-Pros medium. To measure cellular lipid peroxidation, the cells were transferred into medium supplemented with either 100 µM Trolox, 100 µM Dfx, 100 nM Fer-1, 200 nM Lpx-1 or without any addition, containing 2 µM BODIPY 581/591 C11 (BODIPY; 10 mM stock solution in DMSO) and incubated for 2 hr. For the comparison of MitoTracker Red and PI staining, the cells were incubated for 0–4 hr in medium ± Trolox, treated with MitoTracker, as described for the fluorescence microscopy, or for 5 min with 5 µg/ml PI (1 mg/ml stock solution in water) in PBS. For the MitoSOX experiments, the cells were pre-loaded for 10 min with 5 µM of the dye (5 mM stock solution in DMSO) in medium + Trolox and then incubated in medium ± Trolox. As control, the cells were treated for 5 min with 20 ng/ml DAPI (50 µg/ml in water) in PBS. For comparing general and mitochondrial lipid peroxidation, the cells were incubated in medium ± Trolox containing 2 µM BODIPY or 100 nM Mito-PerOx (2 mM stock solution in DMSO).

As proof-of-principle for the experiments with RPA, the cells were transferred into medium supplemented with either 100 µM Trolox, 100 µM Dfx, 100 nM Fer-1 or Trolox plus 100 µM Fe(III)/HQ (freshly prepared by mixing equal volumes of 10 mM $FeCl_3$ in water and 10 mM HQ in 50% v/v ethanol in water; [*Petrat et al., 2002*]) and incubated for 2 hr. Thereafter, the cells were washed, re-suspended in PBS, incubated for 15 min with 50 nM RPA (1 mM stock solution in DMSO), washed and incubated for another 15 min in medium + Trolox to allow optimal enrichment of the dye in the mitochondrion. To follow changes in RPA fluorescence after Trolox withdrawal, starting with the longest time point, the cells were stained with 50 nM RPA or 1 nM RPAC (1 mM stock solution in DMSO) and treated as described above followed by up to 3 hr incubation in medium ± Trolox. To detect possible changes in the mitochondrial iron content due to SODA-overexpression, the Px I-III KO/SODA-myc cells were cultured for about 20 hr ± Tet, and stained with 50 nM RPA. After the respective treatments, the cells were washed with cold PBS, re-suspended in 1 ml cold PBS, transferred into FACS tubes (Sarstedt) and immediately subjected to flow cytometry in a BD FACSCanto or FACSCantoII instrument at the Flow Cytometry and FACS Core Facility (FFCF) of the Center of Molecular Biology (ZMBH) of Heidelberg University. The following excitation lasers and emission filters (ex:em) were applied: BODIPY and MitoPerOx 488:530/30 nm, MitoTracker 561:586/15 nm, PI 488:570 nm, DAPI 405:450/50 nm, MitoSOX 488:585/42 nm, RPA and RPAC 561:610/20. In each experiment 10000 events were recorded. The data were analyzed using FlowJo software (FlowJo, LLC).

## Fluorescence microscopy

About $1.2 \times 10^6$ cells were used per sample. MitoTracker staining, cell fixation, permeabilization and antibody treatment were performed as described previously (*Hiller et al., 2014*; *Schaffroth et al., 2016*). Antibodies against mtTXNPx, VDAC, cMyc and cTXNPx were diluted 1:1000, 1:500, 1:200, and 1:1000, respectively. For visualization, Alexa Fluor 488-conjugated goat anti-guinea pig (1:1000), anti-rabbit (1:1000), Alexa Fluor 546 goat anti-rabbit (1:1000) and Alexa

Fluor 488-conjungated goat anti-mouse (1:250) (Molecular Probes) antibodies were used. The cells were examined using a Carl Zeiss Axiovert 200 M microscope equipped with an AxioCam MRm digital camera and the AxioVision software (Zeiss, Jena, Germany).

## Transmission electron microscopy

Aliquots of $5 \times 10^7$ logarithmically growing Px I-III KO cells were transferred into medium ± Trolox and incubated for 0.5, 1 and 2 hr. The cell fixation and embedding procedures were adapted from (*Höög et al., 2010*). Briefly, cells were fixed by 2.5% glutaraldehyde and 2% p-formaldehyde in 100 mM cacodylate buffer, pH 7.2 at 4°C overnight. The cells were centrifuged in fixative and the pellet was processed in one piece. After rinsing in buffer the samples were further fixed in 1% osmium tetroxide in cacodylate buffer, washed in water, and incubated with 1% uranyl acetate in water overnight. Dehydration was done in 10 min steps in an aceton gradient followed by Spurr resin embedding and polymerization at 60°C. The blocks were cut in 70 nm thin sections using a Leica UC6 ultramicrotome (Leica Microsystems Vienna, Austria) and collected on pioloform-coated mesh grids. The post-stained sections were imaged on a JEOL JEM-1400 electron microscope (JEOL, Tokyo, Japan) operating at 80 kV and equipped with a 4K TemCam F416 (Tietz Video and Image Processing Systems GmBH, Gautig, Germany). The analysis was performed by the Electron Microscopy Core Facility (EMCF) of Heidelberg University.

## Measurement of cellular ATP

Logarithmically growing Px I-III KO cells in SDM-79 or MEM-Pros medium were divided into six aliquots. Starting with the latest time point, the samples were washed with PBS, transferred into 5 ml medium ± Trolox and incubated at 27°C. After 0 to 4 hr, an aliquot was removed and kept on ice for cell counting, another one was treated with PI and subjected to flow cytometry as described above. The remaining cells were centrifuged and re-suspended in 300 μl medium + Trolox. Three aliquots of 100 μl, each containing $2 \times 10^6$ cells, were transferred into a 96-well plate, mixed with 50 μl ATPlite one-step solution (PerkinElmer) according to the manufacturer's protocol and luminescence was measured in a Victor X4 plate reader (PerkinElmer). The luminescence of the cell-free medium was subtracted. All cells that still had an elongated shape or only highly motile cells were counted. All data were analyzed and are presented as percentage of the starting values using GraphPad Prism.

## Western blot analysis of SODA-overexpressing cells

PC Px I-III KO/SODA-myc cells, cultivated for 18 hr in the presence or absence of 1 μg/ml tet, were harvested, re-suspended in reducing SDS-sample buffer and boiled. Lysates from $5 \times 10^6$ cells were loaded per lane onto a 12% gel and subjected to SDS-PAGE. After electrophoresis, the proteins were transferred onto a PVDF membrane and probed with antibodies against c-Myc (1:200) and aldolase (1:20,000), followed by development with HRP-conjugated goat antibodies against mouse (1:5000) and rabbit (1:10,000) IgGs, respectively.

## Live cell imaging

All sample preparations were performed at 27°C in the dark in a total volume of 1 ml. Per sample, about $10^6$ cells were harvested and washed with cold PBS. Treatment of the cells with MitoSOX or MitoPerOx was done as described above for the flow cytometry. MitoSOX-treated cells were then transferred into medium ± Trolox and incubated for 2 hr. Afterwards the cells were treated for 15 min with 120 nM MitoTracker Green in Trolox-supplemented medium, followed by 30 min incubation in Trolox-containing medium for optimal enrichment of the dye in the mitochondrion. For nuclear and kinetoplast DNA staining, the cells were treated for 15 min with 3 μg/ml Hoechst 33342 in PBS. In the case of RPA and RPAC, the cells were incubated for 15 min with 150 nM RPA or 10 nM RPAC in Trolox-supplemented PBS followed by 15 min in Trolox-supplemented medium. Subsequently, the cells were either stained with MitoTracker Green or transferred into medium ± Trolox and incubated for up to 3 hr at 27°C in the dark. After the respective treatments, the cells were washed with PBS, re-suspended in 20 μl PBS, placed on ice and within ≤30 min inspected under a Carl Zeiss Axiovert 200 M microscope equipped with an AxioCam MRm digital camera and the AxioVision software (Zeiss, Jena).

## Biological data collection and data evaluation

Except where stated otherwise, all experiments were performed three times on separate days as independent biological replicates. The data shown represent the mean ± SD of these replicates. The data were evaluated using GraphPad Prism (GraphPad Software, La Jolla, CA).

## Acknowledgements

We are grateful to Dr. Mike Murphy for a generous gift of MitoPerOx and to Drs José Pedro Friedmann Angeli and Marcus Conrad for the RSL3 samples. Drs Christine Clayton, Rafael Radi and André Schneider kindly provided us with antisera. We thank Dr. Monika Langlotz from the Flow Cytometry and FACS Core Facility (FFCF) of the Center of Molecular Biology (ZMBH) of Heidelberg University for helpful advice. Drs Charlotta Funaya, Steffi Gold and Stefan Hillmer from the Electron Microscopy Core Facility (EMCF) of Heidelberg University are acknowledged for the EM analysis and discussions of the results. We thank Natalie Dirdjaja, Torsten Schmenger and Felix Lindheimer for their contributions to the RSL3 studies. This work was supported by the Deutsche Forschungsgemeinschaft (DFG Kr 1242/5–1 and 8–1).

## Additional information

### Funding

| Funder | Grant reference number | Author |
| --- | --- | --- |
| Deutsche Forschungsgemeinschaft | Kr1241/5-1 | R Luise Krauth-Siegel |
| Deutsche Forschungsgemeinschaft | Kr1241/8-1 | R Luise Krauth-Siegel |

The funders had no role in study design, data collection and interpretation, or the decision to submit the work for publication.

### Author contributions

Marta Bogacz, Formal analysis, Validation, Investigation, Visualization, Writing—original draft; R Luise Krauth-Siegel, Conceptualization, Resources, Supervision, Funding acquisition, Writing—original draft, Writing—review and editing

### Author ORCIDs

R Luise Krauth-Siegel https://orcid.org/0000-0003-2164-8116

### Decision letter and Author response

Decision letter https://doi.org/10.7554/eLife.37503.018
Author response https://doi.org/10.7554/eLife.37503.019

## Additional files

### Supplementary files

• Supplementary file 1. Trypanocidal activity and in vitro inhibitory potency of RSL3. (A) Bloodstream *T. brucei* were cultured for 1, 2 or 3 days in the presence of 100 μM Trolox, 200 nM Liproxstatin-1 or 100 nM Ferrostatin-1 as well as different RSL3 concentrations and then subjected to plate reader-based ATPlite measurements. Chlorhexidine, a trypanocidal compound and known inhibitor of trypanothione reductase (TR) (*Meiering et al., 2005*; *Beig et al., 2015*), served as positive control. [1]The data are the mean of at least two independent series of experiments each conducted in triplicate with standard deviations (SD). [2]The values are the mean of an experiment conducted in triplicate with SD. (B) NADPH, $T(SH)_2$, TR, and Tpx ± Px were incubated with 40 μM RSL3. After different times, the assays were started by adding (a) 100 μM $H_2O_2$ or (b) Px and $H_2O_2$. The data were derived from at least double determinations which varied by $\leq$ 10%.

DOI: https://doi.org/10.7554/eLife.37503.015
• Transparent reporting form
DOI: https://doi.org/10.7554/eLife.37503.016

**Data availability**

All data generated or analysed during this study are included in the manuscript and supporting files.

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
