## [Decision Letter]

Thank you for submitting your article "Tryparedoxin peroxidase-deficiency commits trypanosomes to ferroptosis-type cell death" for consideration by eLife. Your article has been reviewed by three peer reviewers, and the evaluation has been overseen by a Reviewing Editor and Wendy Garrett as the Senior Editor. The following individuals involved in review of your submission have agreed to reveal their identity: Scott J Dixon (Reviewer #2); Brent R Stockwell (Reviewer #3).

The reviewers have discussed the reviews with one another and the Reviewing Editor has drafted this decision to help you prepare a revised submission.

The overall consensus is that this is a very timely and interesting manuscript study providing first evidence that loss of the whole tandem gene locus of *T. brucei* tryparadoxin peroxidases I-III (Px, the closest orthologues of mammalian glutathione preroxidase-4, GPx4) causes cell death in the procyclic stage. Notably, this form of cell death displays many features of a novel form of iron-dependent cell death, called ferroptosis, first described in the mammalian system by inactivation of the key ferroptosis regulator GPx4. Hence, it seems that ferroptosis can occur in evolutionarily distant organisms, and that ferroptosis inducing agents may be useful therapeutics for targeting these trypanosomes. Since these trypanosomes are the cause of African sleeping sickness and Nagana cattle disease, this could have relevant implications for human and cattle health.

In general, this is a timely and quite important paper on ferroptosis in a distantly related organism. The manuscript is presented in a straightforward and logical fashion, and the quality of the experiments is sound and robust. Yet the reviewers have raised some concerns about the specificity of certain inhibitors and probes, the potential involvement of mitochondria in the ferroptotic death process and the conceptual advancement beyond on what has been published by the same group earlier (Schaffroth et al., 2016). Nonetheless, the key message conveyed by this study is that ferroptosis is a highly conserved, "ancient" cell death mechanism which besides in humans can also occur in highly specialized unicellular organisms, such as trypanosomes.

Essential revisions:

1) The implication of mitochondria in ferroptosis is still highly controversial and debated in the mammalian system. Indeed, a fair bit of evidence suggests that mitochondrial ROS and mitochondrial morphological alterations are neither necessary nor sufficient for the execution of ferroptosis in mammalian cells. For example, mammalian cells lacking a functional electron transport chain show no differences in susceptibility to ferroptosis compared with wild-type cells (e.g. see Dixon et al., 2012); oxylipidomic results from the Kagan group indicate that the primary mitochondrial phospholipid – cardiolipin – is not oxidized in mammalian cells undergoing ferroptosis; and recent results from the Stockwell lab show that biochemical elimination of all mitochondria from the mammalian cell do not prevent the execution of ferroptosis, and slow this process only slightly. Therefore, a careful and thorough discussion is needed. For instance, is the crucial function of mitochondria in trypanosome ferroptosis (if so) a conserved phenomenon, or is it a specialized property in trypanosomes? There is a single mitochondrion in the unicellular trypanosome, so mitochondrial function and regulation of metabolism in the organelle are probably highly distinctive/specialized in trypanosomes, compared to that in other organisms.

2) The authors use a number of redox-sensitive probes and antioxidants which always harbor the risk to lead to over-interpretations (or even wrong conclusions). For instance, MitoSOX is not necessarily specific for mitochondrial ROS and should either be validated as such or used as a general ROS probe. This is particularly relevant as Figure 4 suggests that MitoSOX has a protective effect against cell death, most likely by acting as a radical scavenger itself. Since interpretation of these data are confounded by the fact that MitoSOX itself prevents the cell from dying, would it not make sense to employ a different probe of mitochondrial ROS that might be independent of such confounding effects? Similarly, the mitochondrial specific localization of MitoPerOx needs to be confirmed.

3) It further seems that there are clear species-specific differences when it comes to the death inducing activity of RSL3. For instance, in Supplementary file 1 the author studied the effect of RSL3 on bloodstream *T. brucei*. The result show that RSL3 can ablate *T. brucei* viability, and RSL3 racemate with Trolox have even lower EC_50_ compared to RSL3 racemate alone. Based on these results the authors conclude that the trypanocidal activity of RSL3 is not due to the inhibition of Px I-III. Does it mean that trypanocidal activity of RSL3 is not through ferroptosis? It thus seems that the lack of selenocysteine in Px renders these enzymes clearly more resistant to RSL3 induced inactivation as shown for mammalian GPx4. Testing whether more selective inhibitors such as ferrostatin-1 and liproxstatin-1 can inhibit the trypanocidal activity of RSL3 will be helpful. In an in vitro system the authors actually confirmed that RSL3 is not able to inhibit Px, but they found that RSL3 inhibited Tpx. So the question is, would Tpx deficiency induce ferroptosis in BS *T. brucei*, or is Tpx inhibition the reason behind the trypanocidal activity of RSL3? Some further clarification would certainly help to improve the current manuscript.

4) The data in Figure 5 on ATP depletion are consistent with the primarily mitochondrial lethal mechanism of action. However, ATP depletion is not a hallmark of ferroptosis in other systems (e.g. see Dixon et al., 2012; Ingold et al., 2018). Along the same line, in Figures 1 and 7 the authors quantified normal vs. dead cells based on certain morphological features (e.g. elongated cells as viable cells, and otherwise dead cells) and ATP-based measurements, which might not be the preferred methods. More direct methods for the measurement of cell death, such as PI (propidium iodide) staining, would be appropriate and should be included.

5) In Figure 8, why were different amounts of the mitochondrial iron chelator (RPA) and control compound (RPAC) used – 150 nM versus 10 nM? It is suggested they demonstrate a “clear mitochondrial localization", but no co-localization data is shown with a bona fide mitochondrial marker. Critically, does RPA prevent the decrease in cell density seen with other inhibitors? The descriptions in the text (subsection “Mitochondrial iron plays a crucial role in the cell death of Px I-III-deficient parasites”) focus on changes in probe fluorescence and cell size from FSC measurements, but the key effect on cell viability (e.g. like in Figure 1A) is not directly reported. If the general model is correct, RPA should behave like Dfx or ferrostatin-1 in the viability assay and completely prevent cell death, one would think. In Figure 8A, it is not clear why different numbers of cells and different time points are shown for different treatment conditions. This figure also needs legends to explain what the two different views of the representative images represent.

6) In Figure 7, the authors claim the specific expression of SODA in the mitochondria, which is hard to judge from the current figures.

7) There are a number of grammatical issues and inaccuracies that should be checked and addressed through careful reading and editing.

– What is the evidence that ferroptosis is involved in "carcinogenesis" (e.g. cancer initiation)?

– "Consequences in the inactivation of GPx4" needs to be re-phrased.

– Introduction: ferroptosis-like, not ferroptosis-type.

– Subsection “Px I-III-deficient cells generate mitochondrial oxidants and lose ATP”, first paragraph: as a 'proof' not 'prove' of principle.

– Discussion, fifth paragraph: "Does not" or should this be "does", since the paper referenced shows that inhibiting all LOX enzymes has an effect?

– "increased DAPI fluorescence": is it DAPI that is being examined here? Or is it PI?

– The sentence (subsection “The Px I-III-deficient cells encounter mitochondria-specific membrane damages”, end of last paragraph) that the cell death may assume features of necroptosis (Stockwell et al., 2017) is not correct. It is not clear what in the 2017 Stockwell review on ferroptosis this is referring to. Is this supposed to say 'necroptosis' or ferroptosis? In case for necroptosis, can necroptosis inhibitors mitigate Px I-III deficiency-induced cell death?

---

## [Author Response]

Essential revisions:1) The implication of mitochondria in ferroptosis is still highly controversial and debated in the mammalian system. Indeed, a fair bit of evidence suggests that mitochondrial ROS and mitochondrial morphological alterations are neither necessary nor sufficient for the execution of ferroptosis in mammalian cells. For example, mammalian cells lacking a functional electron transport chain show no differences in susceptibility to ferroptosis compared with wild-type cells (e.g. see Dixon et al., 2012); oxylipidomic results from the Kagan group indicate that the primary mitochondrial phospholipid – cardiolipin – is not oxidized in mammalian cells undergoing ferroptosis; and recent results from the Stockwell lab show that biochemical elimination of all mitochondria from the mammalian cell do not prevent the execution of ferroptosis, and slow this process only slightly. Therefore, a careful and thorough discussion is needed. For instance, is the crucial function of mitochondria in trypanosome ferroptosis (if so) a conserved phenomenon, or is it a specialized property in trypanosomes? There is a single mitochondrion in the unicellular trypanosome, so mitochondrial function and regulation of metabolism in the organelle are probably highly distinctive/specialized in trypanosomes, compared to that in other organisms.

We are aware of the fact that the role of mitochondria in ferroptosis is still debated and that an active respiratory chain is not a required for the death program. We also cite the work by Kagan et al. (2017) that cardiolipin peroxidation was not observed in Pfa-1 cells subjected to ferroptosis but also the work of Friedmann-Angeli et al. (2104) who showed that GPx4-deficient kidneys display time-dependent formation of peroxidized cardiolipin. Our data strongly suggest that the individual cell type largely determines the site where ferroptosis starts. This is especially obvious in the trypanosomal system. The mitochondrial phenotype displayed by the peroxidase-deficient procyclic *T. brucei* described here is not a specialized property of trypanosomes and their single mitochondrion. In bloodstream parasites – which also have a single mitochondrion but, in contrast to the insect form, rely exclusively on glycolysis, lack an active mitochondrial respiratory chain and take up iron via receptor-mediated endocytosis of host transferrin – the lysosomal membrane is the primary site affected when the cytosolic peroxidases are missing (Hiller et al. 2016). We restructured and expanded the Discussion and included additional references to clarify these points.

2) The authors use a number of redox-sensitive probes and antioxidants which always harbor the risk to lead to over-interpretations (or even wrong conclusions). For instance, MitoSOX is not necessarily specific for mitochondrial ROS and should either be validated as such or used as a general ROS probe. This is particularly relevant as Figure 4 suggests that MitoSOX has a protective effect against cell death, most likely by acting as a radical scavenger itself. Since interpretation of these data are confounded by the fact that MitoSOX itself prevents the cell from dying, would it not make sense to employ a different probe of mitochondrial ROS that might be independent of such confounding effects? Similarly, the mitochondrial specific localization of MitoPerOx needs to be confirmed.

We fully agree that the subcellular localization of mitochondria-targeted sensors is not restricted to the mitochondrion. However, e. g. MitoSOX has been shown to be highly enriched in the mitochondrion (Jauslin et al., 2003; Robinson et al., 2008) which suggests that also a putative protecting activity is more pronounced in the mitochondrion. To our knowledge, virtually all commercially available small molecule sensors for reactive oxygen species irreversibly react with the metabolite(s) they sense. This should lower the effective concentration of the reactive species and thus lead to a partial protection. We are currently cloning vectors to express roGFP-based sensors in the cytosol and mitochondrion of the parasites to overcome this problem. We studied MitoSOX and MitoPerOx-treated parasites by life cell imaging with MitoTracker Green as mitochondrial marker and Hoechst 33342 to visualize nuclear and mitochondrial DNA. As long as the cells retained their normal morphology, MitoSOX visualized a single small dot which coincided with the Hoechst signal for the kinetoplast and thus binding of the sensor to the mitochondrial DNA. A nuclear DNA staining was not observed (new Figure 4A). MitoPerOx revealed a perfect overlay with MitoTracker Green (new Figure 6A). The data strongly suggest that both sensors are highly enriched in the mitochondrion. The text has been modified accordingly.

3) It further seems that there are clear species-specific differences when it comes to the death inducing activity of RSL3. For instance, in Supplementary file 1 the author studied the effect of RSL3 on bloodstream T. brucei. The result show that RSL3 can ablate T. brucei viability, and RSL3 racemate with Trolox have even lower EC_50_ compared to RSL3 racemate alone. Based on these results the authors conclude that the trypanocidal activity of RSL3 is not due to the inhibition of Px I-III. Does it mean that trypanocidal activity of RSL3 is not through ferroptosis? It thus seems that the lack of selenocysteine in Px renders these enzymes clearly more resistant to RSL3 induced inactivation as shown for mammalian GPx4. Testing whether more selective inhibitors such as ferrostatin-1 and liproxstatin-1 can inhibit the trypanocidal activity of RSL3 will be helpful. In an in vitro system the authors actually confirmed that RSL3 is not able to inhibit Px, but they found that RSL3 inhibited Tpx. So the question is, would Tpx deficiency induce ferroptosis in BS T. brucei, or is Tpx inhibition the reason behind the trypanocidal activity of RSL3? Some further clarification would certainly help to improve the current manuscript.

Our data, indeed, indicate that RSL3 lethality of the parasites is not through ferroptosis as the trypanocidal activity of RSL3 was unaffected in the presence of Trolox, and, as shown in the revised Supplementary file 1, also of Fer-1 and Lpx-1. All three radical-trapping antioxidants failed to prevent the cytotoxicity of RSL3 strongly suggesting that the peroxidases are not the targets. This lack of sensitivity may at least partially be explained by the fact that the parasite enzymes have an active site cysteine. As shown by Ingold et al. (2018), replacement of the selenocysteine-containing GPx4 by a cysteine mutant renders mammalian cells resistant towards RSL3. Our in vitro assays showed that RSL3 inhibits Tpx. This distant relative of thioredoxins and glutaredoxins is the general and essential oxidoreductase that transfers reducing equivalents from trypanothione not only to the peroxidases but also methionine-sulfoxide reductase and, probably most importantly, ribonucleotide reductase. Thus, inhibition of Tpx should affect the synthesis of DNA precursors which could be the main reason for the trypanocidal action of RSL3. We included the new data in Supplementary file 1 and modified the text to improve and clarify this section.

4) The data in Figure 5 on ATP depletion are consistent with the primarily mitochondrial lethal mechanism of action. However, ATP depletion is not a hallmark of ferroptosis in other systems (e.g. see Dixon et al., 2012; Ingold et al., 2018). Along the same line, in Figures 1 and 7 the authors quantified normal vs. dead cells based on certain morphological features (e.g. elongated cells as viable cells, and otherwise dead cells) and ATP-based measurements, which might not be the preferred methods. More direct methods for the measurement of cell death, such as PI (propidium iodide) staining, would be appropriate and should be included.

Figure 2— figure supplement 1 provides PI staining and shows that in the absence of a protecting agent loss of the mitochondrial membrane potential precedes cell death. We now studied the ATP levels also in comparison to PI staining. The data corroborated that ATP depletion precedes damage of the cell membrane (new Figure 5B). To corroborate the cell counting data for the SODA-overexpressing cells we also studied the cells by flow cytometry after PI treatment. The results confirmed that overexpression of SODA or a low concentration of Dfx slows down cell lysis and that in combination the Px I-III KO cells are more efficiently protected. In the absence or presence of Dfx, overexpression of SODA improved the cell viability (new Figure 7D).

5) In Figure 8, why were different amounts of the mitochondrial iron chelator (RPA) and control compound (RPAC) used – 150 nM versus 10 nM? It is suggested they demonstrate a “clear mitochondrial localization", but no co-localization data is shown with a bona fide mitochondrial marker. Critically, does RPA prevent the decrease in cell density seen with other inhibitors? The descriptions in the text (subsection “Mitochondrial iron plays a crucial role in the cell death of Px I-III-deficient parasites”) focus on changes in probe fluorescence and cell size from FSC measurements, but the key effect on cell viability (e.g. like in Figure 1A) is not directly reported. If the general model is correct, RPA should behave like Dfx or ferrostatin-1 in the viability assay and completely prevent cell death, one would think. In Figure 8A, it is not clear why different numbers of cells and different time points are shown for different treatment conditions. This figure also needs legends to explain what the two different views of the representative images represent.

The reason for the use of different concentrations of the compounds is that RPAC-treated cells displayed a much higher fluorescence than those loaded with RPA. In the fluorescence microscopy, at the shortest exposure time, the signal for 150 nM RPAC was already oversaturated. To follow a putative increase or quenching of the fluorescence by flow cytometry, again different concentrations of RPA and RPAC had to be applied. To show that RPA, but not RPAC, had a protecting effect we used the compounds at an equal concentration of 150 nM because only light scattering of the cells was studied. We modified the text to make this point clearer. We co-stained the RPA- and RPAC-treated cells with MitoTracker Green which confirmed the mitochondrial localization of the compounds (new Figure 8A). We also followed the viability of the peroxidase-deficient cells after treating them with RPA and show that pre-loading with RPA slowed down cell lysis (new Figure 8C).

6) In Figure 7, the authors claim the specific expression of SODA in the mitochondria, which is hard to judge from the current figures.

We improved the IF pictures to make the co-localization of SODA and MitoTracker and the lack of overlay with a cytosolic marker protein clearer.

7) There are a number of grammatical issues and inaccuracies that should be checked and addressed through careful reading and editing.– What is the evidence that ferroptosis is involved in "carcinogenesis" (e.g. cancer initiation)?

The expression “carcinogenesis” was used by Stockwell et al. (2017). Because it may be misleading we replaced it by “cancer”.

– "Consequences in the inactivation of GPx4" needs to be re-phrased.

We re-phrased the sentence.

– Introduction: ferroptosis-like, not ferroptosis-type.

We replaced “type” by “like”.

– Subsection “Px I-III-deficient cells generate mitochondrial oxidants and lose ATP”, first paragraph: as a 'proof' not 'prove' of principle.

We corrected the typing error.

– Discussion, fifth paragraph: "Does not" or should this be "does", since the paper referenced shows that inhibiting all LOX enzymes has an effect?

The text is correct. Yang et al. (2016) show that inhibiting all LOX enzymes has an effect if ferroptosis is induced by glutathione depletion but not if ferroptosis is caused by inactivation of GPx4. Nevertheless we modified the text to make the point clearer.

– "increased DAPI fluorescence": is it DAPI that is being examined here? Or is it PI?

The text is correct. In the experiment, DAPI was used because the MitoSOX fluorescence would interfere with that of PI.

– The sentence (subsection “The Px I-III-deficient cells encounter mitochondria-specific membrane damages”, end of last paragraph) that the cell death may assume features of necroptosis (Stockwell et al., 2017) is not correct. It is not clear what in the 2017 Stockwell review on ferroptosis this is referring to. Is this supposed to say 'necroptosis' or ferroptosis? In case for necroptosis, can necroptosis inhibitors mitigate Px I-III deficiency-induced cell death?

We removed the last sentence.